# Multiscale Deep Spatial Feature Extraction Using Virtual RGB Image for Hyperspectral Imagery Classification

**Liqin Liu** [1,2,3], **Zhenwei Shi** [1,2,3,*], **Bin Pan** [4], **Ning Zhang** [5], **Huanlin Luo** [5] and **Xianchao Lan** [5]

1. Image Processing Center, School of Astronautics, Beihang University, Beijing 100191, China; liuliqin@buaa.edu.cn
2. Beijing Key Laboratory of Digital Media, Beihang University, Beijing 100191, China
3. State Key Laboratory of Virtual Reality Technology and Systems, School of Astronautics, Beihang University, Beijing 100191, China
4. School of Statistics and Data Science, Nankai University, Tianjin 300071, China; panbin@buaa.edu.cn
5. Shanghai Aerospace Electronic Technology Institute, Shanghai 201109, China; max15@buaa.edu.cn (N.Z.); hlluo18@fudan.edu.cn (H.L.); senlei@buaa.edu.cn (X.L.)
* Correspondence: shizhenwei@buaa.edu.cn

**Abstract:** In recent years, deep learning technology has been widely used in the field of hyperspectral image classification and achieved good performance. However, deep learning networks need a large amount of training samples, which conflicts with the limited labeled samples of hyperspectral images. Traditional deep networks usually construct each pixel as a subject, ignoring the integrity of the hyperspectral data and the methods based on feature extraction are likely to lose the edge information which plays a crucial role in the pixel-level classification. To overcome the limit of annotation samples, we propose a new three-channel image build method (virtual RGB image) by which the trained networks on natural images are used to extract the spatial features. Through the trained network, the hyperspectral data are disposed as a whole. Meanwhile, we propose a multiscale feature fusion method to combine both the detailed and semantic characteristics, thus promoting the accuracy of classification. Experiments show that the proposed method can achieve ideal results better than the state-of-art methods. In addition, the virtual RGB image can be extended to other hyperspectral processing methods that need to use three-channel images.

**Keywords:** hyperspectral image classification; feature extraction; fully convolutional networks (FCN); virtual RGB image; multiscale spatial feature

## 1. Introduction

The rapid development of remote sensing technology in recent years has opened a door for people to more profoundly understand the earth. With the development of imaging technology, hyperspectral remote sensing has become one of the most important directions in the field of remote sensing. Because of their rich spectral information, hyperspectral images have been widely used in environmental monitoring, precision agriculture, smart city, information defense, resource management and other fields [1–3]. Hyperspectral classification is an important research branch of hyperspectral image processing, which assigns each pixel its corresponding ground category label [4].

Since the sample labeling of hyperspectral images is very difficult, how to use finite samples to obtain higher classification accuracy becomes the main problem in the field of hyperspectral classification [5]. Researchers have conducted in-depth research on this issue. At present there are mainly two directions, one is to extract more expressive features from hyperspectral images [6],

and the other is to design better classifiers [7]. In terms of feature extraction, the feature of adding spatial information performs better in the classification. Therefore, the commonly used feature is the spatial–spectral fusion feature. In 2005, Benediktsson et al. proposed a method based on extended morphology combined with spatial information, which is the earliest known method of combining spatial and spectral features [8]. Afterwards, many scholars have expanded and proposed more hyperspectral classification algorithms based on the joint of space and spectrum, such as [9–11] et al. For the extraction of spatial features, scholars have adopted a variety of ways. Li et al. proposed loopy belief propagation for classification [12], Jia et al. put forward a Gabor filter for feature extraction [13], Pan et al. raised hierarchical guidance filtering to use a spatial feature [14]. On the design of classifiers, the most classic is the support vector machine (SVM) [15–17]. Based on SVM, Gu et al. proposed a kernel-based learning method, which combines various kernel functions in a weighted joint instead of a single kernel to improve classification performance [18,19]. In addition, many other classifiers are also applied on hyperspectral images, such as the semi-supervised method based on graphics (LNP), the sparse representation based classifier (SRC), random forest (RF) and extreme learning machine (ELM) [20–24]. Traditional machine learning hyperspectral classification methods solved many problems in hyperspectral classification and can realize the classification of different species. However, the methods cannot meet the requirement of accuracy on the condition that samples is very limited. In recent years, deep learning technologies have been widely used in image processing, the layer of networks is deepening [25–29], the tasks to solve are more and more varied [30–34]. Scholars gradually choose to apply deep networks solving the problem of hyperspectral classification. Benefiting from the expression ability of deep features, the deep learning based methods obtain better classification performance. However, due to the large number of samples required for deep learning which contradicts the limited number of hyperspectral classification samples, there are still problems for us to solve.

The deep learning method stacked auto-encoders (SAE) was applied on hyperspectral classification by Chen in 2014 [35]. It is essentially a five-layer structure for extracting deep information from original features. After then, a convolutional neural network (CNN) is widely used on hyperspectral image classification [36–40]. With the development of various deep neural networks, they are used to solve the hyperspectral classification tasks [41–45]. For example, Paoletti and Wang respectively proposed capsule networks on classification, Paoletti used the hyperspectral patches into the capsule network and Wang separately extracts the information in spatial and spectral domains [41,42]. Zhong used deep residual networks (DRN) [29] to construct a 3D framework to classify the hyperspectral data [43]. Mou and Zhu put forward recurrent neural network (RNN) and generative adversarial networks (GAN) [31] on hyperspectral classification, respectively [44,45]. Actually, instead of processing the hyperspectral image itself as an image, the above methods constructed each pixel as an image or a classification object. That is to say, the networks are proposed to classify the features of each pixel as an object. These networks artificially construct a data volume for each pixel to be classified as an input. This input can be seen as the simplest shallow features and the method actually turned into a process to obtain the deep features from the input shallow features.

On account of the fact that CNN cannot get rid of the manual raw features construct, a fully convolutional network (FCN) is introduced into the hyperspectral classification. The goal of an FCN is to achieve pixel level segmentation [46], which is consistent with the task of hyperspectral classification. An FCN does not construct features for each pixel that needs to be classified artificially, thus, it is more suitable for the hyperspectral classification task. Jiao used an FCN for hyperspectral image classification by inputting the dimension-reduced images to the trained model on natural images [47]. The method used the spatial features linear mapped to the 20 classes of natural images, which will cause information loss. Niu et al. extracted features using DeepLab [48], evaluated the discriminative ability of the weighted fusion features by the visualization method, without further utilization and analysis of spatial features. Li et al. proposed a convolution–deconvolution neural network [49], by forming images of the first principal component and each band, generating multiple

groups of images for network training and realizing the classification of hyperspectral images. In this method, the connection between the bands of each spectrum is ignored, which may cause loss of spectral information.

Aiming at the above problems, this paper proposes a multiscale deep spatial feature extraction using a virtual RGB image method (MDSFV). Hyperspectral images are constructed into virtual RGB images which makes the distribution of its color features more similar to the natural images used in training network. The image is fed into the trained FCN model and multiscale features are extracted. By combining these features in a skip-layer way, the semantic information in deep features and edge and detail information in shallow features will be taken into account at the same time, which is more conducive to the pixel-level classification of hyperspectral images. In this process, we adopt a layer-by-layer normalization combining, in order to balance information from different layers. Finally, spatial spectral fusion features are sent into the classifier for classification to obtain the final objects' distribution.

The main contributions of this paper are summarized as follows:

1.  A new three-channel image is constructed to overcome the input limitation of FCN models. The hyperspectral bands corresponding to RGB wavelength are selected out. By simulating the Gaussian effect of photographic sensing on the RGB band, we introduced Gaussian weights to combine the corresponding bands. Compared with the simple three-dimensional extraction of the principal component, virtual RGB image is more suitable for the needs of the trained network, and hopefully extracts more useful features. In addition to benefiting the depth model feature extraction in this paper, this method can also be widely applied to any hyperspectral processing algorithm that needs to construct three-channel images.
2.  Based on fully convolution networks, a multi-layer feature fusion method is proposed. Compared with the previous methods in which the spatial feature extraction is directly based on the feature fusion of the network itself, the proposed method directly operates on the features of different layers thus reduce the loss of classify-related information. After the multi-layer spatial features extracted by FCN, features of different scales are combined via upsampling, cropping, combining, etc. Deep features will provide more semantic information, which is more beneficial to the discrimination of categories, shallow features will provide more edge and detail information, which facilitate to the expression of contour information of ground objects. The cross-layer jointing can preserve both sementic and detail information, making the feature more expressive.
3.  For the characteristics of different layers, different dimensions and semantic scales, a new joint of features is applied. After the scales of the features are unified, instead of simply concatenating the features, the method unifies their dimensions by extracting the principal components and then the different layer features are normalized and added together. The principal component analysis (PCA) based feature change can retain as much information as possible. Then the features are normalized and added layer-by-layer. Combining their corresponding layers directly can avoid the drawbacks of the feature dimension increasing caused by directly concatenate them, is more conducive to accurate and rapid classification.

The rest of this paper is organized as follows. In Section 2, we give a detailed description about the proposed method. In Section 3, experiments on three popular datasets are provided. In Section 4, we analyze the parameters involved in the algorithm. We conclude this paper in Section 5.

## 2. Materials and Methods

The labeled hyperspectral image data is very limited. In addition, the imaging conditions of different hyperspectral images, the number of spectral bands and the ground objects are significantly different, which make different hyperspectral data that cannot be trained together like natural images and other remote sensing images. The fully convolutional network can be used to classify hyperspectral images because its task is to perform pixel-level class determination of the entire image, which is

consistent with the goal of achieving classification of hyperspectral images. There are many parameters of the FCN, thus the single hyperspectral image cannot complete the update of all network parameters. By constructing a three-channel virtual RGB image, this paper simulates the trained network model on natural images for the pixel-level segmentation process, and better adapts to the characteristics of existing models. In this way, multi-layer and multiscale spatial features are extracted, and multiscale features joint is realized through various feature processing techniques, which enhances the feature expression ability. Finally, the spatial and spectral features are fused together to realize the classification of hyperspectral images. The procedure of the method is shown in Figure 1, mainly concluding three-channel image construction, multi-layer multiscale feature extraction and jointing, the process of the feature fusion and classification.

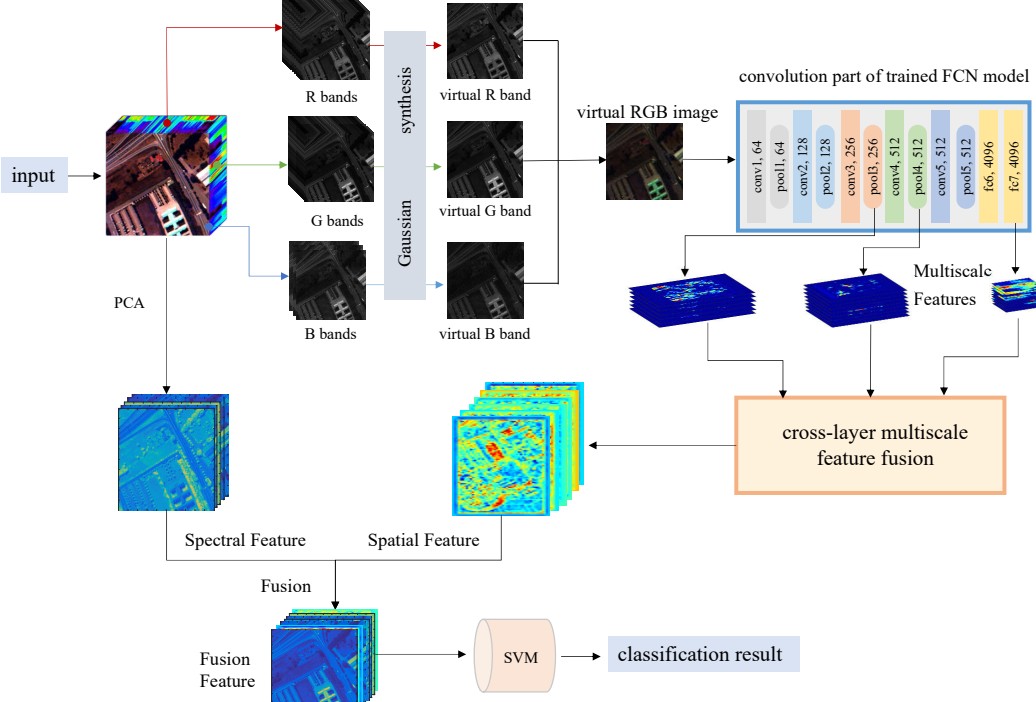

**Figure 1.** The procedure of the multiscale deep spatial feature extraction using a virtual RGB image method (MDSFV). The corresponding bands of RGB wavelength are selected and combined into a virtual RGB image, then the images are fed into the trained fully convolutional network (FCN) model to extract the multiscale features. The blue box shows the structure of the FCN convolution section and the orange box shows the skip-layer feature fusion section which is detailed in the next figure. The multiscale features are joined to obtain the spatial feature and the spectral feature is fused in the last for classification.

## 2.1. Virtual RGB Image Construction

Hyperspectral imaging spectrometers can form approximately continuous spectral curves for tens or even hundreds of bands, including red, green and blue bands of visible light and some near-infrared bands. In recent years, many scholars have used the well-trained networks on natural images to extract features from hyperspectral images, such as CNN [50] and FCN [47]. The common way is to perform PCA on the whole spectrum of hyperspectral images, and then select the first three principal components to form a three-channel image into the networks [47,49], detailed process can be find in [47]. In this way, the difference between the bands of hyperspectral images vanishes, and the advantages of wide spectral range and narrow imaging band may be lost. Here, considering that the existing model is trained on the natural image of RGB (three-band), it may be more suitable for the spatial feature extraction of RGB corresponding wavelength bands. So we construct a virtual

RGB image to make full use of the hyperspectral's rich band information through applying it to the extraction of spatial features using existing networks. The specific implementation is as follows:

**a.** According to the information of the imaging spectrometer, the wavelength bands corresponding to the wavelengths of red, green and blue, respectively, are selected from the hyperspectral image. That is, the red band corresponds to the wavelength range 625–750 nm, the green bands corresponds wavelength 495–570 nm and the blue bands represents wavelength 435–450 nm. Since the luminance signal generated by the photosensitive device is not generated by single-band illumination, here we simply establish a photosensitive model simulating its synthesis process in the RGB bands.

**b.** The gray value of each band of the natural image is modeled as Gaussian synthesis of all narrow bands in the red, green and blue bands. Take R band as example, suppose the band contains $b$ spectral bands, and the corresponding hyperspectral band tabs are $c_1, c_2, \ldots, c_b$ and the band reflection value is $s_1, s_2, \ldots, s_b$. Then we think that the distribution of the bands conforms to the $3\sigma$ principle, that is, the mean and variance of the Gaussian distribution are

$$\begin{aligned} \mu_R &= (c_1 + c_b)/2 \\ \sigma_R &= (c_b - c_1)/6. \end{aligned} \tag{1}$$

Then we can get the weight of the band $s_k$ is

$$f(s_k) = \frac{1}{\sqrt{2\pi}\sigma_R} \exp -\frac{(c_k - \mu_R)^2}{2(\sigma_R)^2}. \tag{2}$$

The resulting synthesized band reflection value is

$$I_R = \frac{\sum_{i=1}^{b} s_k f(s_k)}{\sum_{i=1}^{b} f(s_k)}. \tag{3}$$

**c.** In order to ensure that the gray value range of the natural image is consistent, the R, G and B gray values of all the pixels are adjusted to the range 0–255.

$$I_R(x,y) = 256 \times \frac{I_R(x,y) - \min_{a,b} I_R(a,b)}{\max_{a,b} I_R(a,b) - \min_{a,b} I_R(a,b)} - 1 \tag{4}$$

Similarly, we can get the equivalent gray value of G band and B band.

So far, we have obtained a virtual RGB image that simulates the RGB image, which will be used as the basis for spatial feature extraction.

*2.2. Spatial Feature Extraction and Skip-Layer Jointing*

The method acquires pixel-wise spatial features by deep and shallow feature fusion. Here we propose a method for extracting spatial features from models trained on natural images for hyperspectral classification. We select FCN for feature extraction. The advantage of FCN is that it has the same target with the hyperspectral image classification, aiming at pixel-wise classification. We reasonably guess that compared to a CNN, features from the FCN are more useful. We applied a well-trained network on natural images to extract multi-layer, multiscale features. Shallow features contain more edge and detailed information of the image, which is especially important for distinguishing the pixel categories of different objects intersections in hyperspectral images, while deep features contain more abstract semantic information, which is important for the determination of pixel categories. Therefore, we extracted both shallow edge texture information and deep semantic structure information, and combined them to obtain more expressive features. We selected VGG16 to

extract spatial features from the virtual RGB images. The parameters were transferred from the FCN trained on ImageNet. In Figure 1, the blue box shows the structure of the convolution part.

During the pooling operation of the fully convolutional network (FCN), the down-sampling multiples of spatial features increased gradually and the semantic properties of features were more and more abstract. The fc7 layer provided semantic information and the shallower layers provided more detailed information. So we chose both deep and shallow features, we extracted the detailed features of the pool3, pool4 and fc7 layers and combined them.The down-sampling multiples from the original image were 8, 16 and 32 times, respectively. We used a layer-by-layer upsampling skip-layer joint to combine the extracted features of the three layers and obtained the final spatial features.

The joint is shown in Figure 2. The method mainly through the upsampling, cropping and two layer feature maps joint to realize layer-by-layer joint expecting improve the ability of expressing spatial characteristics. It can better preserve the depth of semantic information in the fc7 layer, and simultaneously combine the edge texture information of the shallow features to improve the ability of feature expression. Since the FCN adds a surrounding zero padding operation to ensure the full utilization of the edge information during the convolution process, mismatch of the feature map and the edge of the original image is caused. Therefore, we focused on the number of pixels that differ between the different layer feature maps when combining skip-layer features. It seriously affects the correspondence between different pixel information of each feature map, which is very important for the pixel-level hyperspectral classification task. The edge pixels of the feature map corresponding outside the reference map are usually defined as offset, and as is known to all, when passing the pooling layer, the offset halves, and when through the convolution layer, the offset caused is

$$S_{off} = S_{pad} - (S_{kernel} - 1)/2. \tag{5}$$

The process is mainly divided into the three operations: upsampling, cropping and the deep features skip-layer joint. The upsampling is mainly based on bilinear interpolation, and in order not to lose the edge information, there is a surrounding padding = 1 to the map. The shallow feature maps occurred after the layer-to-layer convolution from the zero padded original image. The original image only corresponds to a part of its center, and the number of pixels in other areas are the offset. So in the cropping, we crop the maps by half of the multiple surrounding them. Since FCN's upsampling and alignment operations are well known to many scholars, we will not cover them further here. We mainly introduce the skip-layer feature map jointing operation.

By cropping, we can guarantee that the two feature maps are equal in size and position, but the dimensions of the two layers are different and cannot be directly added. If the two layers of features are directly concatenated, the feature dimension is multiplied, which greatly affects the computational efficiency and classification performance. Here we used principal component analysis (PCA), reducing the dimension of the feature map with a high dimension to make the dimension of the two maps the same and adding them layer by layer. In order to ensure the spatial information contribution of the two-layer feature is equivalent, each dimension of the two layer feature map was normalized before the addition. In the process of dimensionality reduction, by the characteristics of the PCA itself, our available principal components are less than $w \times h - 1$, $w \times h$ is the size of feature to PCA. So if the dimension of the shallower is $m$ and that of the deeper is $n$ ($m \leq n$), the size of the deeper map is $w \times h$, the dimension of the two maps is $\min(w \times h - 1, m)$.

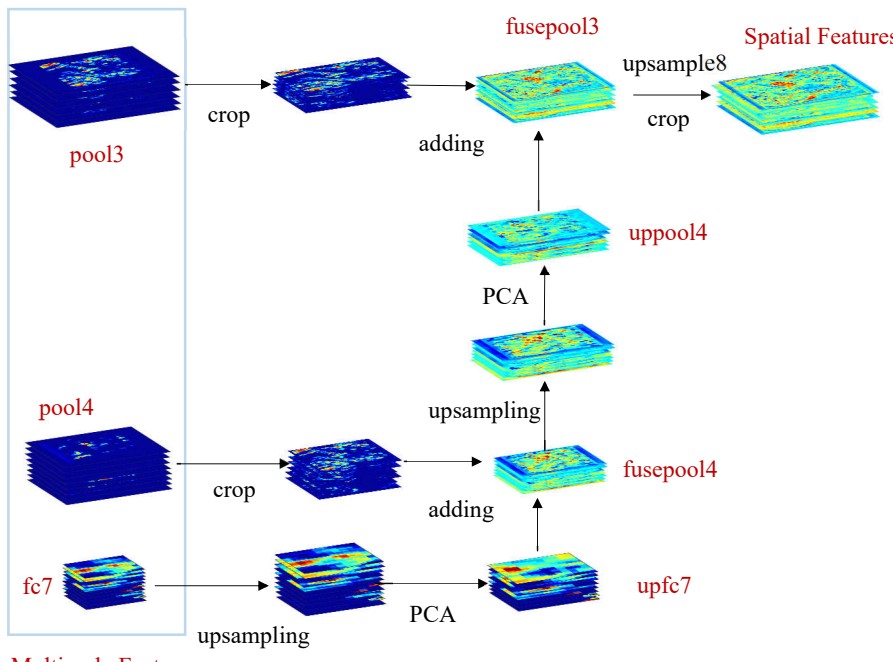

**Figure 2.** Multiscale features to spatial features. The red word represents the name of the feature layers, and the rest represents some processing of the feature map.

Combined with the convolution process of FCN, we conducted the following analyses for the feature joint.

**Offset during the convolution process.**

A series of operations of the FCN in the convolution process were analyzed to obtain the scale relationship and positional association before each feature map. According to (5), the parameters of the convolution between the layers of the FCN are shown in Table 1.

**Table 1.** Detailed configuration of the FCN convolutional part.

| Layer | Kernel | Stride | Padding | dim | Offset Made |
|---|---|---|---|---|---|
| conv1_1 | $3 \times 3$ | 1 | 100 | 64 | 99 |
| conv1_2 | $3 \times 3$ | 1 | 1 | 64 | 0 |
| pool1 | $2 \times 2$ | 2 | 0 | 64 | half |
| conv2_1 | $3 \times 3$ | 1 | 1 | 128 | 0 |
| conv2_2 | $3 \times 3$ | 1 | 1 | 128 | 0 |
| pool2 | $2 \times 2$ | 2 | 0 | 128 | half |
| conv3_1 | $3 \times 3$ | 1 | 1 | 256 | 0 |
| conv3_2 | $3 \times 3$ | 1 | 1 | 256 | 0 |
| conv3_3 | $3 \times 3$ | 1 | 1 | 256 | 0 |
| pool3 | $2 \times 2$ | 2 | 0 | 256 | half |
| conv4_1 | $3 \times 3$ | 1 | 1 | 512 | 0 |
| conv4_2 | $3 \times 3$ | 1 | 1 | 512 | 0 |
| conv4_3 | $3 \times 3$ | 1 | 1 | 512 | 0 |
| pool4 | $2 \times 2$ | 2 | 0 | 512 | half |
| conv5_1 | $3 \times 3$ | 1 | 1 | 512 | 0 |
| conv5_2 | $3 \times 3$ | 1 | 1 | 512 | 0 |
| conv5_3 | $3 \times 3$ | 1 | 1 | 512 | 0 |
| pool5 | $2 \times 2$ | 2 | 0 | 512 | half |
| fc6 | $7 \times 7$ | 1 | 0 | 4096 | -3 |
| fc7 | $1 \times 1$ | 1 | 0 | 4096 | 0 |

In Table 1, we can find the offset of fc7 and the original image is 0, so we used the fc7 layer as the benchmark when combining skip-layer features, and the baseline was selected as the deeper feature map when cropping.

**Offset calculation between feature maps.**

We discuss the calculation of the offset between the two layers of feature maps before and after the pooling and other convolution operations. We assume that the offset of the deep feature relative to the original image is $O_d$, the offset generated by the convolution layer is $O_c$ and during the upsampling padding = 1, the offset is 1, then the offset of a feature layer former relative to the latter layer $O_{ds}$, is calculated as follows:

$$O_{ds} = k \times (O_d - O_c) - k/2 \tag{6}$$

where $k$ represents the downsampling times of the two layers. There is a pooling layer between them, so $k = 2$. The detailed offset between the layers through different feature levels are shown in Table 2.

**Table 2.** Different levels of feature joints and offsets.

| Deep Feature Levels | Feature Name | Crop Offset |
|---|:---:|:---:|
| Two-Layer Joint | fc7 | – |
| | pool4 | 5 |
| | uppool4 (pool4 upsample 16 times) | 27 |
| Three-Layer Joint | fc7 | – |
| | pool4 | 5 |
| | pool3 | 9 |
| | uppool3 (pool3 upsample 8 times) | 31 |
| Four-Layer Joint | fc7 | – |
| | pool4 | 5 |
| | pool3 | 9 |
| | pool2 | 13 |
| | uppool2 (pool2 upsample 4 times) | 35 |

**Unified dimension of deep and shallow feature maps.**

We discuss the number of principal components retained by the deeper feature map after PCA and the dimension of the shallower feature map. We take the feature of the upsampled PCA to reduce the dimension, and finally take the dimension as the minumum of the shallower feature and the dimension of the deeper feature after PCA. If the shallower's dimension is the larger, PCA will be also used on it to change the dimension to that of the deeper feature after PCA.

**The detailed process of skip-layer feature joint is as follows:**

**Joint fc7 and pool4**

First we upsampled the fc7 layer. The size of the fc7 and pool4 layer maps was different. Because of the padding operation, the relative offset between the two layers was generated. From Table 1, we knew that offset between fc7 and the original image is 0. In Table 2, the relative offset between the two layers was 5, the pool4 layer was cropped according to the offset. PCA was performed on the upsampled fc7 layer and we selected the same dimension of the upsampled fc7 and pool4. Then, we added their features layer-by-layer to obtain the feature map after fusion. We named it fuse-pool4 layer, which was offset from the original image by 5.

**Joint fuse-pool4 and pool3**

We combined the fuse-pool4 layer and the pool3 layer in the same way. We upsampled the fuse-pool4 and selected the feature map after upsampling according to the uniform rules of the deep and shallow feature dimensions. The relative offset between the two layers was 9 calculated, as shown in Table 2. The pool3 layer was cropped according to the offset to obtain two layers of the same size. Then they were added layer-by-layer to get the feature map fuse-pool3 after the fusion.

**Upsample to image size**

The fuse-pool3 layer was 8 times downsampled relative to the original image. We applied the upsampling process to directly upsample the layer by 8 times, and then calculated the offset between it and the original image. The offset between the upsampled feature map and the original image was 31 and we cropped the feature map after upsampling according to offset = 31, and the spatial features corresponding to the pixel level of the original image were obtained.

So far, we have obtained the spatial features corresponding to the original image, which combines the features of 8 times, 16 times and 32 times downsampling of deep neural network. They not only include the edge and detail information required for hyperspectral pixel-level classification, but also contains semantic information needed to distinguish pixel categories. The feature map corresponds to the original image as much as possible, which can effectively reduce the possibility of generating classification errors in the two types of handover positions. In the process of deep and shallow feature fusion, we adopt the uniform of the dimension of two layer feature maps, and then add them layer by layer. Compared with directly concatenating the features, the feature dimensions are effectively reduced, and the ability of feature expression of the layers are maintained.

*2.3. Spatial–Spectral Feature Fusion and Classification*

We combined the hyperspectral bands of RGB-corresponding wavelengths to construct a virtual RGB image, and then use FCN to extract multi-layer, multiscale features of the image. Through the skip-layer joint of these features, the spatial features that characterize the spatial peculiarity of the pixel and the surrounding distribution are obtained. However, in the process of extracting the feature, we ignored the other hyperspectral bands and the close relationship between the bands. Therefore, we extracted the spectral features associated with each pixel's spectral curve and fused it with the spatial features for classification.

Because the hyperspectral band is narrow and the sensitization range is wide, the number of hyperspectral bands is huge. To ensure that the feature dimension is not too high during the classification process and the expression ability of the feature is not affected as much as possible, we carried out the spectral curve for PCA. After the PCA, we selected the former masters of the composition as a spectral feature of the pixel.

We combined the spatial and spectral features of the corresponding pixels. For the characteristics of different sources, the common joint was to directly concatenate the different features. Here, we considered that the range of values and the distribution of data were different between the spatial features obtained by the network extraction and the spectral characteristics represented by the spectral reflection values. We normalized the features and combined them according to the equations shown in Equations (8) and (9) to fuse spatial features and spectral features. Suppose $X_{spe}$ is a spectral feature, which is obtained from the original spectrum PCA taking the first $s_e$ principal component, and $X_{spa}$ is the deep spatial feature with a dimension of $s_a$, so we know

$$X_{spe} \in R^{w \times h \times s_e}, X_{spa} \in R^{w \times h \times s_a} \tag{7}$$

where $w \times h$ is the size of the features to fuse. First, we do the following for $X_{spe}, X_{spa}$ to normalize them in Equation (8)

$$\bar{X}_d = \frac{1}{w \times h} \sum_{i=1}^{n} \sum_{j=1}^{n} X_{ij}$$

$$\sigma_d = \frac{1}{w \times h} \sum_{i=1}^{n} \sum_{j=1}^{n} (X_{ij} - \bar{X}_d)^2 \tag{8}$$

$$f(X_{ij}) = (\frac{X_{ij} - \bar{X}_d}{\sigma_d}) / |\frac{X - \bar{X}_d}{\sigma_d}|$$

In Equation (8), We perform the normalization operation of the layers of the spectral and spatial features, that is, subtract the average then divide by the variance operation on the corresponding features of each pixel, so as to achieve the uniformity of each feature dimension. Then we combine features by concatenating them, the size of the fused feature is shown in Equation (9).

$$X_f \in R^{w \times h \times (s_a + s_e)} \tag{9}$$

where $s_e, s_a$ is discussed in Section 4; the occurred $X_f$ is the feature after fusion, we fed it into the classifier to implement classification.

## 3. Results

In this section, we evaluate the performance of the proposed MDSFV by comparing it with some state-of-art methods. The experiments were performed on three datasets, mainly including data from two hyperspectral imaging sensors, the Airborne Visible/Infrared Imaging Spectrometer (AVIRIS) and Reflective Optics System Imaging Spectrometer (ROSIS-3). Meanwhile, we also carried out comparative experiments on different feature joint ways to verify the effectiveness of multiscale skip-layer features and different three-channel image to verify the effectiveness of virtual RGB image.

### 3.1. Data Description and Experiment Setup

We selected the Indian Pines dataset, the Pavia University dataset and the Kennedy Space Center dataset, which we will cover separately.

The Indian Pines dataset was collected in 1992 by the Airborne Visible/Infrared Imaging Spectrometer (AVIRIS) at the test site in northwestern Indiana. The dataset has a spatial dimension of $145 \times 145$ pixels with a spatial resolution of 20 m/pixel and a total of 220 wavelength reflection bands covering a wavelength range of 0.4–2.5 μm with a nominal spectral resolution of 10 nm. After removing the influence band of noise and water absorption ([104–108], [150–163], 220), the remaining 200 bands were used for experiments. The scene is mainly agriculture and forests, including a small number of buildings. The dataset contains a total of 16 classes. The sample size varies greatly among classes, the minimum is only 20 and the largest class has 1428 samples. When selecting training samples, we selected according to the ratio of the total number of samples. In this experiment, we selected each class of 10% for training, and other samples for testing. The PCA first three components image, the virtual RGB image and the corresponding objects label map are as shown in Figure 3.

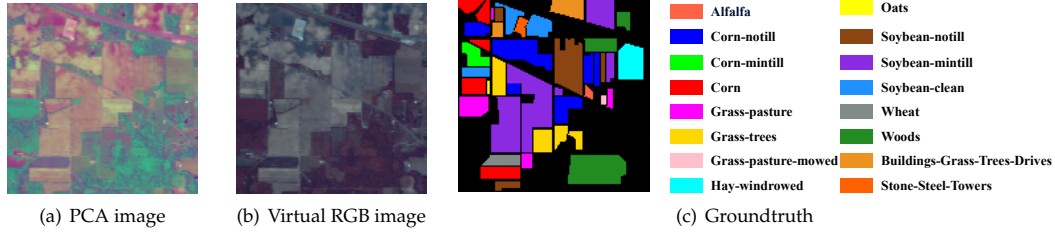

    (a) PCA image      (b) Virtual RGB image      (c) Groundtruth

**Figure 3.** Indian Pines dataset, (**a**) shows the image of first three principal component by principal component analysis (PCA); (**b**) shows the virtual RGB image by our method; the groundtruth is in (**c**).

Pavia University dataset was taken in Pavia, Italy by the Reflective Optics System Imaging Spectrometer (ROSIS-3). The sensor has a total of 115 spectral channels covering a range of 0.43–0.86 µm. After removing noise and water absorption bands, there were 103 hyperspectral bands remaining. The spatial dimension is 610×340, the spatial resolution is 1.3 m/pixel, and a total of 42,776 samples were included, containing nine types of ground objects such as grass, trees and asphalt. Since each type of labeled sample has a large amount, we chose 50 samples from each class to train and all remaining samples to test. In Figure 4, the three components image, the virtual RGB image and label of Pavia University are displayed.

Kennedy Space Center (KSC) dataset was obtained in 1996 by the Airborne Visible/Infrared Imaging Spectrometer (AVIRIS) at the Kennedy Space Center (KSC). A total of 224 bands, covering the wavelength range of 0.4–2.5 µm, the KSC dataset is available at a height of approximately 10 km with a spatial resolution of 18 m/pixel. After removing the water absorption and low SNR (Signal Noise Ratio) bands, there are 176 bands for analysis, including 13 objects coverage categories. The categories of the dataset are relatively balanced and the number is several hundred, so we used 20 samples of each type for training. Figure 5 shows the three-channel image by PCA, the virtual RGB image and the label map of KSC.

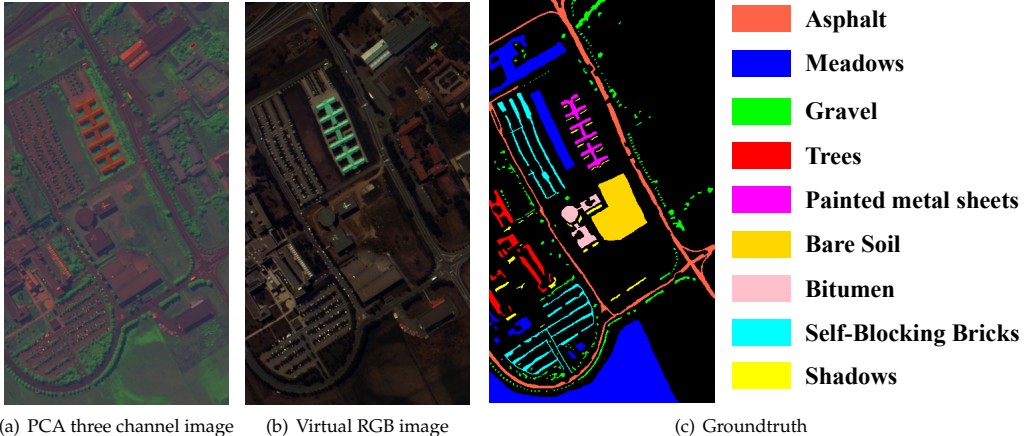

(a) PCA three channel image　　　(b) Virtual RGB image　　　　　　　　(c) Groundtruth

**Figure 4.** Pavia University dataset, (**a**) shows the first three components; (**b**) shows the virtual RGB image by our method, the groundtruth is in (**c**).

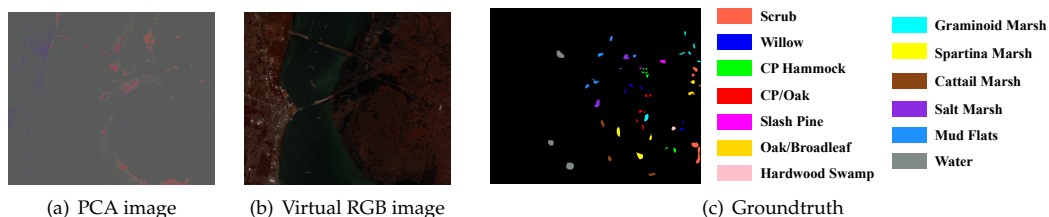

(a) PCA image　　　　　　(b) Virtual RGB image　　　　　　　(c) Groundtruth

**Figure 5.** Kennedy Space Center dataset, (**a**) is the PCA's first three channels; (**b**) shows the virtual RGB image we proposed; (**c**) shows the groundtruth.

Our experiments were based on the fully convolutional network of the caffe framework to extract spatial features [51], using Matlab to configure LIBSVM (A Library for Support Vector Machines) by Chih-Chung Chang and Chih-Jen Lin [52] and LIBLINEAR (A Library for Large Linear Classification) [53] for feature fusion and classification calculations. Under the same conditions, we compare the proposed MDSFV method with the method only using the deepest features upsampling it 32 times and without changing dimension (DSFV). Simultaneously, four CNN or FCN based methods are chosen to compare, including DMS³FE [47] and FEFCN [49] based on FCN and two CNN [50] and CNN PPF [54] based on CNN. All the above methods were run 10 times with randomly selected train or test samples, and the average accuracies and the corresponding standard deviations are reported.

We selected overall accuracy (OA), average accuracy (AA) and kappa coefficient ($\kappa$) to evaluate the performance of these methods, and the accuracies of each class shown in the tables are calculated by the proportion of correctly classified samples to total test samples.

### 3.2. Feature Jointing and Fusion Strategies

In this section, we will compare the effectiveness of our strategy in the feature joint and fusion, including the depth of the spatial feature acquired from FCN, and report on how the deeper and shallower combined and the efficiency of the fusion of spectral-spatial feature. We selected the Pavia University dataset to compare the results of each strategy.

**The Depth of the Spatial Feature**

In this section, we show the results of the joint between the different feature layers extracted by FCN. From the convolution process of FCN, we know that the downsampling multiples of pool2, pool3, pool4 and fc7 layers relative to the original image are 4, 8, 16 and 32, respectively. Here we compare the three levels' feature sets, the features used by each are shown in Table 2. For example, the joint of Two-Layer is to combine the fc7 layer feature with the pool4 layer feature, and is upsampled 16 times to obtain the spatial feature map corresponding to the original image. Shallow features will make the classification more detailed, deep features will carry more semantic information; the question of how to balance the relationship between them to obtain accurate and precise classification results is a problem.

Figure 6 shows the classification results of three levels. We can see that the Four-Layer method has some subtle misclassifications due to the joint of the characteristics of the shallow pool2. The relative reduction of shallow information in the Three-Layer method improves these misclassifications. When the shallow feature is further reduced, the Two-Layer method is adopted, since the feature image is directly sampled by 16 times, it is inaccurate in the range of 16 pixels, thus causing some other misclassifications. Table 3 demonstrates the accuracy of each class of the three levels' joint, we can further observe the advantages of the Three-Layer method. Combining the above results to balance the relationship between the deep and shallow layers, we use the joint of the Three-Layer deep spatial features.

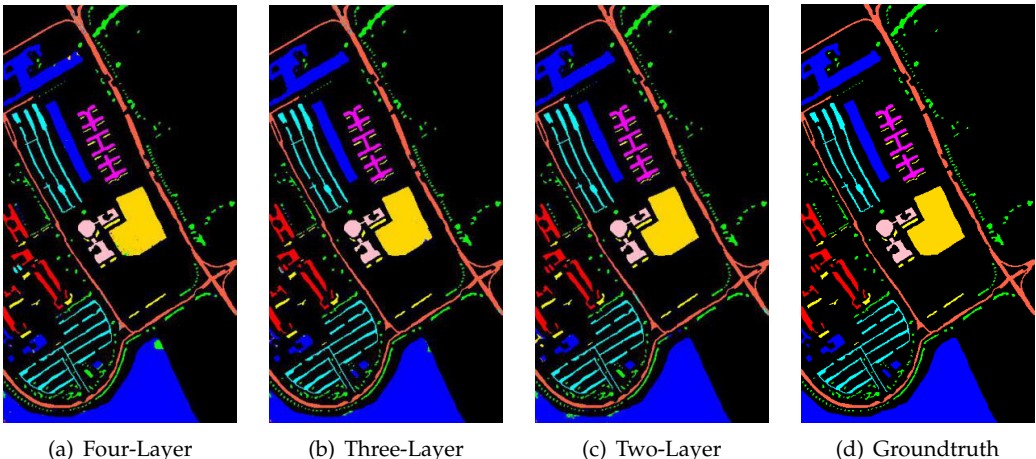

　　(a) Four-Layer　　　　　(b) Three-Layer　　　　　(c) Two-Layer　　　　　(d) Groundtruth

**Figure 6.** Results of different layer feature joints on the Pavia University dataset.

**Table 3.** Accuracy of different levels of feature joints on the Pavia University dataset.

| Class | Train Sample | Test Sample | Two-Layer | Three-Layer | Four-Layer |
|-------|--------------|-------------|-----------|-------------|------------|
| 1 | 100 | 6531 | 0.9821 | 0.9884 | **0.9891** |
| 2 | 100 | 18549 | 0.9831 | **0.9943** | 0.9846 |
| 3 | 100 | 1999 | 0.957 | 0.9885 | **0.9985** |
| 4 | 100 | 2964 | 0.9642 | **0.9835** | 0.9703 |
| 5 | 100 | 1245 | 0.9984 | **1** | **1** |
| 6 | 100 | 4929 | 0.986 | 0.9923 | **0.9992** |
| 7 | 100 | 1230 | 0.9911 | **0.9992** | **0.9992** |
| 8 | 100 | 3582 | 0.9925 | 0.9947 | **0.9972** |
| 9 | 100 | 847 | **1** | **1** | **1** |
| overall accuracy(OA) | | | 0.9825 | **0.9925** | 0.989 |
| average accuracy(AA) | | | 0.9838 | **0.9934** | 0.9931 |
| kappa($\kappa$) | | | 0.9768 | **0.9901** | 0.9853 |

**The Combination of Deeper and Shallower Features**

In the method section of this paper, we mentioned that for different layer feature maps, the dimension-unified features are normalized and added. In this part, we compare the results of three combination ways, which are: the different layer features are directly concatenated (Concat); applying PCA to features of the same dimension and then adding them layer-by-layer (No Normalization); and the PCA after features are normalized before combining (Normalization).

Table 4 demonstrated the accuracy and time-consumption of the three combination ways. When the two layer features are merged in the concatenation, the final feature dimension is very large, reaching 4896 dimensions, resulting in significant time-consumption in the process of upsampling bilinear interpolation. The memory footprint is more than 32 GB. Thus, we decided not to choose this combination mode when the classification result is equivalent. The main comparison is whether the normalization has an effect on the accuracy when the training samples are identical. We can see that the normalization addition is, relatively, a little inferior to the non-normalization direct addition on time consumption. In terms of classification effect, the normalization addition is better than the non-normalized. In Section 2.2 we also analyzed this method, because the method is implemented to make the contribution of two layers' information the same, so we chose the method of normalization and addition. Figure 7 shows the results with and without normalization.

**Table 4.** Accuracy of the different combined methods on the Pavia University dataset.

| Class | Train Sample | Test Sample | Concat | No Normalization | Normalization |
|-------|--------------|-------------|--------|------------------|---------------|
| 1 | 100 | 6531 | **0.9781** | 0.9675 | 0.9633 |
| 2 | 100 | 18549 | 0.9812 | 0.9752 | **0.9834** |
| 3 | 100 | 1999 | **1** | 0.9985 | 0.9985 |
| 4 | 100 | 2964 | 0.9615 | **0.9899** | 0.9879 |
| 5 | 100 | 1245 | 0.9992 | 0.9992 | **1** |
| 6 | 100 | 4929 | **1** | **1** | 0.9996 |
| 7 | 100 | 1230 | **1** | **1** | 0.9992 |
| 8 | 100 | 3582 | 0.9956 | **0.9961** | 0.9894 |
| 9 | 100 | 847 | 0.9889 | 0.9988 | **1** |
| overall accuracy(OA) | | | **0.985** | 0.9828 | **0.985** |
| average accuracy(AA) | | | 0.9894 | **0.9917** | 0.9912 |
| kappa($\kappa$) | | | **0.9801** | 0.9772 | **0.9801** |
| time consume | | | $\gg$5 h | 14.50 s | 14.90 s |

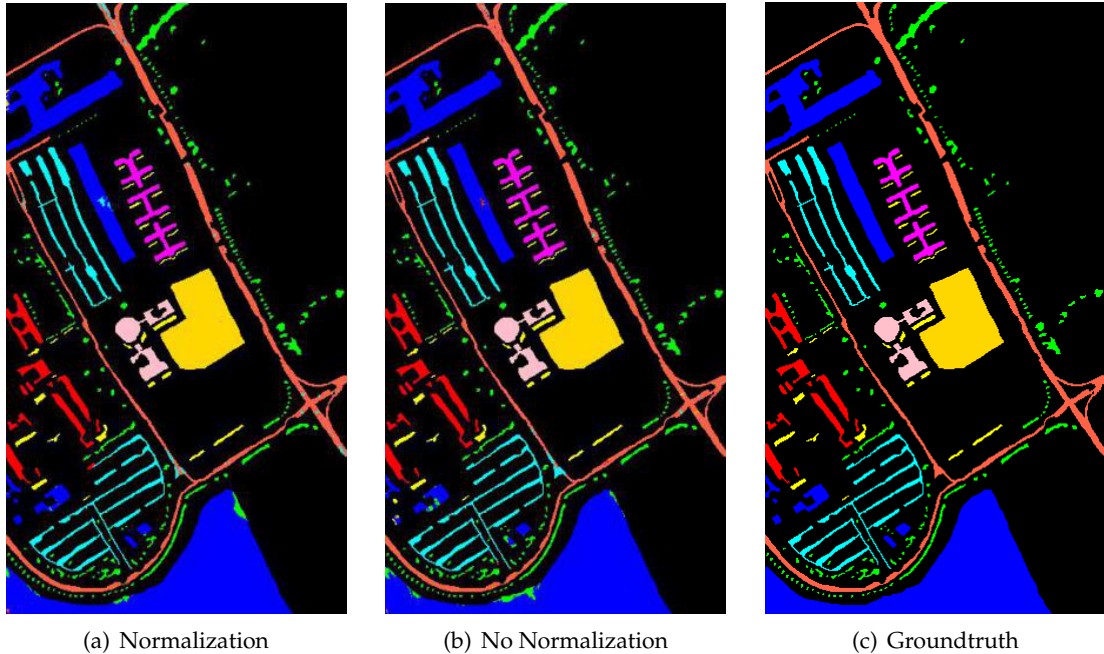

(a) Normalization       (b) No Normalization       (c) Groundtruth

**Figure 7.** Efficiency of normalization on the Pavia University dataset.

**Fusion of Spectral and Spatial Feature**

We know that spatial and spectral features are both important for accurate classification of pixels. Spatial features provide information about the neighborhood around the pixel. Spectral features provide unique, discerning and accurate spectral curves that are unique to the pixel. Spatial information is conducive to the continuity of pixel classification. Spectral information is important to the accuracy of specific pixel classification, so if you want to accurately classify, spatial and spectral features are indispensable. This section will compare the classification results using only spectral features, using only spatial features and fusion spatial and spectral features.

Figure 8 exhibits the results of using spectral or feature, respectively, and that of the fusion feature. When using the spectral feature only, the continuity of the spatial distribution of ground features is affected—pixels in the shadow class are classified very scattered. Nevertheless, when the spatial feature is the only uniquely usable feature, even if the classification result is better in the area of the piece, in the slenderly distributed area, especially the road, the result is not ideal (divided into sections). This can be interpreted as the result of missing spectral information only by using spatial information. When applying spatial information and spectral information at the same time, we can find that the classification result is better, while avoiding the scattered distribution of pixel classes and slender object mistakes. The phenomenon of segmentation is basically close to the groundtruth. Table 5 gives the accuracies of each species in three ways. The high accuracy is shown in bold, and it can be seen that the effectiveness of the spatial–spectral fusion is very obvious.

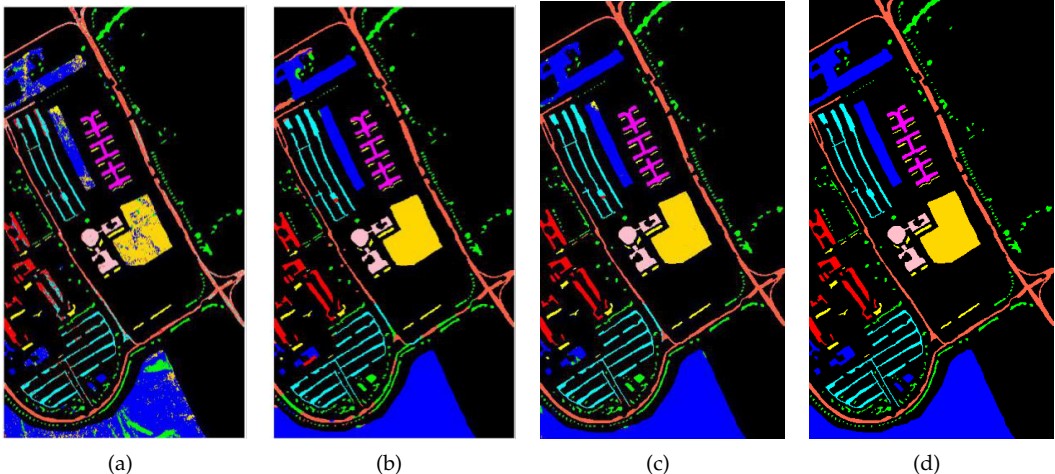

|     |     |     |     |
| --- | --- | --- | --- |
| (a) | (b) | (c) | (d) |

**Figure 8.** Result of the spectral spatial feature fusion on the Pavia University dataset. (**a**,**b**) show the classification results of spectral and spatial features, respectively. (**c**) shows the result of spectral–spatial fusion feature and (**d**) displays the groundtruth.

**Table 5.** Efficiency of the spatial spectral feature fusion on the Pavia University dataset.

| Class | Train Sample | Test Sample | Spectral | Spatial | Fusion |
| --- | --- | --- | --- | --- | --- |
| 1 | 100 | 6531 | 0.7809 | 0.8738 | **0.9778** |
| 2 | 100 | 18,549 | 0.821 | 0.9541 | **0.9781** |
| 3 | 100 | 1999 | 0.7809 | 0.99 | **0.998** |
| 4 | 100 | 2964 | 0.9497 | 0.9109 | **0.9919** |
| 5 | 100 | 1245 | **0.9992** | 0.9904 | **0.9992** |
| 6 | 100 | 4929 | 0.8643 | **1** | 0.9988 |
| 7 | 100 | 1230 | 0.935 | **1** | 0.9992 |
| 8 | 100 | 3582 | 0.8227 | 0.9701 | **0.983** |
| 9 | 100 | 847 | **0.9965** | 0.9481 | 0.9728 |
| overall accuracy(OA) | | | 0.8394 | 0.9493 | **0.9840** |
| average accuracy(AA) | | | 0.8833 | 0.9597 | **0.9887** |
| kappa($\kappa$) | | | 0.7975 | 0.9334 | **0.9787** |

### 3.3. Effectiveness of Virtual RGB Image

In this part, we compare the performance of different three-channel images. PCA the hyperspectral data directly, the average of RGB corresponding bands and Gaussian combination of the bands are compared. For PCA, we select the first three principal components as the RGB channel intensity. The other two methods select the bands have same wavelength range with the RGB, and combine them in different methods. One is averaging the bands (average bands) and another is synthesizing the bands by Gaussian weights (virtual RGB).

In Table 6 and Figure 9, we can see that the RGB corresponding bands can achieve a better classification result than the PCA method, which is because the RGB bands are more similar to the natural images training model. Since the Gaussian weights performance is better on this account, we selected the virtual RGB images for feature extraction.

**Table 6.** Accuracy of different three-channel images on the Pavia University dataset.

| Class | Train Sample | Test Sample | PCA | Average Bands | Virtual RGB |
|-------|--------------|-------------|-----|---------------|-------------|
| 1 | 50 | 6581 | 0.9809 | 0.9672 | **0.9829** |
| 2 | 50 | 18599 | 0.9592 | 0.9526 | **0.9714** |
| 3 | 50 | 2049 | 0.998 | **0.9985** | **0.9985** |
| 4 | 50 | 3014 | 0.9879 | 0.9437 | **0.9889** |
| 5 | 50 | 1295 | **1** | **1** | **1** |
| 6 | 50 | 4979 | 0.999 | 0.9982 | **0.9992** |
| 7 | 50 | 1280 | **0.9992** | 0.9984 | **0.9992** |
| 8 | 50 | 3632 | **0.993** | 0.9913 | 0.9899 |
| 9 | 50 | 897 | **1** | **1** | **1** |
| | overall accuracy(OA) | | 0.9772 | 0.9688 | **0.9828** |
| | average accuracy(AA) | | 0.9908 | 0.9833 | **0.9922** |
| | kappa($\kappa$) | | 0.97 | 0.959 | **0.9772** |

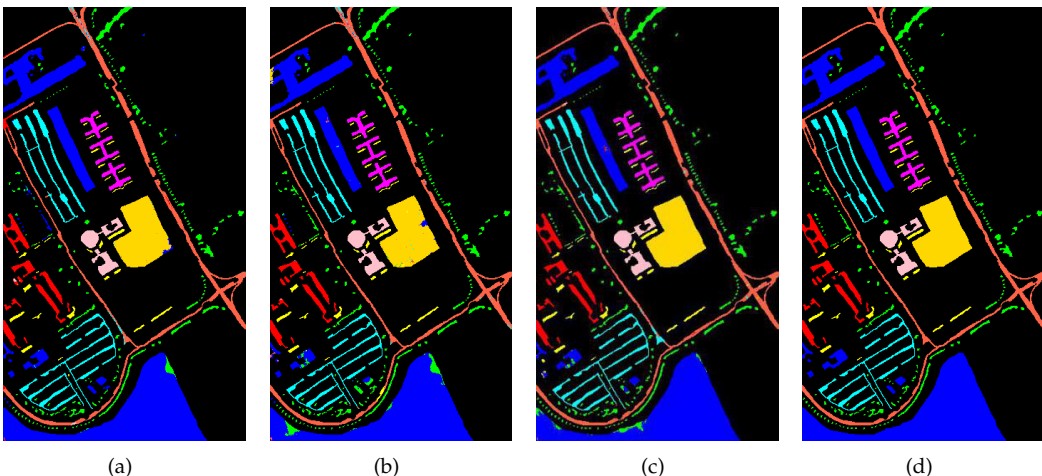

|        |        |        |        |
|:------:|:------:|:------:|:------:|
| (a) | (b) | (c) | (d) |

**Figure 9.** Result of different three-channel images on the Pavia University dataset. (**a**) shows the classification result of PCA three channel image. (**b**) shows the results of average bands method. (**c**) shows the result of virtual RGB image and (**d**) displays the groundtruth.

### 3.4. Classification Performance

In this section, we will compare the proposed MDSFV method with spatial feature extraction without multiscale features' skip-layer fusion (DSFV) method as baseline and the other four state-of-art methods. The result figures and detailed average accuracy and variance are shown in Figures 10–12 and Tables 7–9.

Figure 10 and Table 7 show the classification performance of the six methods on Indian Pines, the experiment were repeated ten times. Figure 10 demonstrates the classification result of each pixel, we can see that of the MDSFV method is closer to groundtruth, and the rest of the methods have more or less mis-segmentations of some regional edges. Table 7 shows the average accuracy and standard deviation per class. Distinctly, our methods are more optimal in terms of various class accuracies. Since CNN PPF selects three different spectral curves in one class and pairs them with other classes, the number of training samples in the ninth class should be 2, so we set the number of samples of our ninth class 3. When comparing with Two CNN method, we did not train and migrate the model on similar datasets. Therefore, the accuracy of this method is significantly lower than the other methods, which also confirms that the training of deep learning networks using hyperspectral data needs a data foundation. The impact of the reduced sample size on the accuracy is enormous. In general, our method achieved an overall accuracy of 98.78%, and DSFV refers to the classification result without skip-layer feature fusion. It can be seen that using multiscale features can improve the recognition accuracy. The comparison to other methods shows it is superior to other deep learning methods.

**Table 7.** Accuracies of different methods on the Indian Pines dataset (%).

| Class | Train Sample | Test Sample | Two CNN | CNN PPF | FEFCN | DMSFE | DSFV | MDSFV |
|---|---|---|---|---|---|---|---|---|
| 1 | 5 | 41 | 69.76 ± 24.54 | 92.93 ± 3.53 | 66.19 ± 9.74 | 92.2 ± 5.09 | 94.39 ± 6.9 | **96.83 ± 2.2** |
| 2 | 143 | 1285 | 54.34 ± 6.18 | 92.46 ± 1.12 | 91.67 ± 1.89 | 96.68 ± 1.47 | 75.26 ± 11.73 | **98.71 ± 0.58** |
| 3 | 83 | 747 | 71.47 ± 2.7 | 86.09 ± 3.61 | 90.46 ± 3.73 | 97.86 ± 1.16 | 89.83 ± 6.72 | **98.25 ± 1.2** |
| 4 | 24 | 213 | 70.14 ± 6.33 | 93.52 ± 2.65 | **98.51 ± 1.16** | 93.9 ± 5 | 96.9 ± 1.99 | 97.93 ± 1.93 |
| 5 | 48 | 435 | 78.32 ± 2.96 | 93.84 ± 1.32 | 90.52 ± 3.96 | **96.71 ± 1.84** | 94.9 ± 2.74 | 96.21 ± 2.54 |
| 6 | 73 | 657 | 86.58 ± 7.37 | 98.66 ± 0.99 | 97.56 ± 1.7 | 99.38 ± 0.56 | 99.53 ± 0.49 | **99.91 ± 0.14** |
| 7 | 3 | 25 | 54.4 ± 14.44 | **89.2 ± 6.94** | 86.04 ± 7.92 | 84 ± 10.58 | 83.2 ± 21.89 | 87.6 ± 8.85 |
| 8 | 48 | 430 | 99.88 ± 0.12 | 99.67 ± 0.26 | **99.9 ± 0.23** | 99.81 ± 0.23 | 100 ± 0 | 99.81 ± 0.31 |
| 9 | 2 | 18 | 51.11 ± 13.79 | 85.29 ± 4.74 | **90.37 ± 10.3** | 52.22 ± 23.2 | 70.56 ± 25.46 | 54.44 ± 17.88 |
| 10 | 97 | 875 | 80.05 ± 5.92 | 91.99 ± 2.24 | 75 ± 2.43 | 95.5 ± 1.18 | 87.74 ± 8.85 | **98.05 ± 0.96** |
| 11 | 246 | 2209 | 62.74 ± 7.63 | 94.25 ± 0.7 | 89.91 ± 1.28 | 98.74 ± 0.57 | 95.33 ± 5.44 | **99.38 ± 0.3** |
| 12 | 59 | 534 | 76.05 ± 7.44 | 92.53 ± 3.92 | 89.4 ± 3.12 | 97.06 ± 1.65 | 91.03 ± 10.2 | **97.75 ± 0.61** |
| 13 | 21 | 184 | 97.66 ± 2.7 | 98.53 ± 0.91 | 99.36 ± 0.67 | 99.29 ± 0.35 | **99.67 ± 0.36** | 99.51 ± 0.16 |
| 14 | 127 | 1138 | 78.88 ± 5.29 | 96.81 ± 0.82 | 99.68 ± 0.24 | 99.97 ± 0.08 | 99.1 ± 2.34 | **100 ± 0** |
| 15 | 39 | 347 | 30.17 ± 12.7 | 79.14 ± 8.01 | 95.59 ± 1.65 | 98.9 ± 1.11 | **99.25 ± 0.66** | 99.08 ± 1.13 |
| 16 | 9 | 84 | 67.62 ± 10.99 | 97.38 ± 2.37 | 97.34 ± 3.25 | 95.48 ± 4.45 | 88.45 ± 3.69 | **98.33 ± 2.62** |
| | overall accuracy(OA) | | 70.52 ± 3.13 | 93.38 ± 0.87 | 91.07 ± 0.91 | 97.84 ± 0.33 | 92.2 ± 2.61 | **98.78 ± 0.19** |
| | average accuracy(AA) | | 70.57 ± 2.98 | 92.33 ± 1 | 91.09 ± 1.06 | 93.61 ± 2.2 | 91.57 ± 3.04 | **95.11 ± 1.44** |
| | Kappa | | 0.6788 ± 0.0324 | 0.9252 ± 0.0098 | 0.898 ± 0.0104 | 0.9755 ± 0.0038 | 0.9122 ± 0.0289 | **0.9861 ± 0.0021** |

**Table 8.** Accuracies of different methods on the Pavia University dataset (%).

| Class | Train Sample | Test Sample | Two CNN | CNN PPF | FEFCN | DMSFE | DSFV | MDSFV |
|---|---|---|---|---|---|---|---|---|
| 1 | 50 | 6581 | 79.86 ± 4.28 | 95.52 ± 0.42 | 88.69 ± 1.1 | 95.94 ± 1.13 | 92.08 ± 3.72 | **97.17 ± 1.53** |
| 2 | 50 | 18599 | 89.24 ± 3.82 | 84.05 ± 2.72 | 94.8 ± 0.58 | 94.78 ± 3.12 | 92.1 ± 3.37 | **96 ± 1.66** |
| 3 | 50 | 2049 | 65.24 ± 5.87 | 90.35 ± 1.17 | 86.19 ± 2.15 | 99.42 ± 0.69 | **99.92 ± 0.22** | 98.23 ± 0.8 |
| 4 | 50 | 3014 | 95.47 ± 1.9 | 92.36 ± 1.2 | 78.67 ± 3.05 | **97.88 ± 0.68** | 96.49 ± 1.4 | 97.32 ± 0.9 |
| 5 | 50 | 1295 | 98.28 ± 1.78 | 99.93 ± 0.11 | 81.3 ± 1.84 | **99.97 ± 0.04** | 99.97 ± 0.07 | 99.93 ± 0.02 |
| 6 | 50 | 4979 | 56.42 ± 8.75 | 92.46 ± 2.44 | 81.9 ± 2.7 | 98.43 ± 0.44 | 98.9 ± 1.24 | **99.29 ± 0.5** |
| 7 | 50 | 1280 | 73.98 ± 4.15 | 94.41 ± 0.32 | 81.82 ± 2.14 | **99.77 ± 0.06** | 99.76 ± 0.53 | 99.51 ± 0.74 |
| 8 | 50 | 3632 | 40.15 ± 5.78 | 86.16 ± 1.8 | 68.28 ± 2.2 | **99.25 ± 0.09** | 97.81 ± 1.73 | 98.71 ± 0.76 |
| 9 | 50 | 897 | 99.68 ± 0.34 | 99.04 ± 0.49 | 86.93 ± 3.61 | 99.59 ± 0.68 | 98.81 ± 0.94 | **99.86 ± 0.26** |
| | overall accuracy(OA) | | 79.03 ± 1.87 | 89.02 ± 1.28 | 86.68 ± 0.33 | 96.63 ± 1.36 | 94.69 ± 1.59 | **97.31 ± 0.79** |
| | average accuracy(AA) | | 77.59 ± 1.59 | 92.7 ± 0.56 | 83.18 ± 0.76 | 98.34 ± 0.39 | 97.31 ± 0.61 | **98.44 ± 0.27** |
| | Kappa | | 0.7344 ± 0.0219 | 0.8608 ± 0.0152 | 0.8241 ± 0.0044 | 0.956 ± 0.0175 | 0.9312 ± 0.02 | **0.9647 ± 0.0102** |

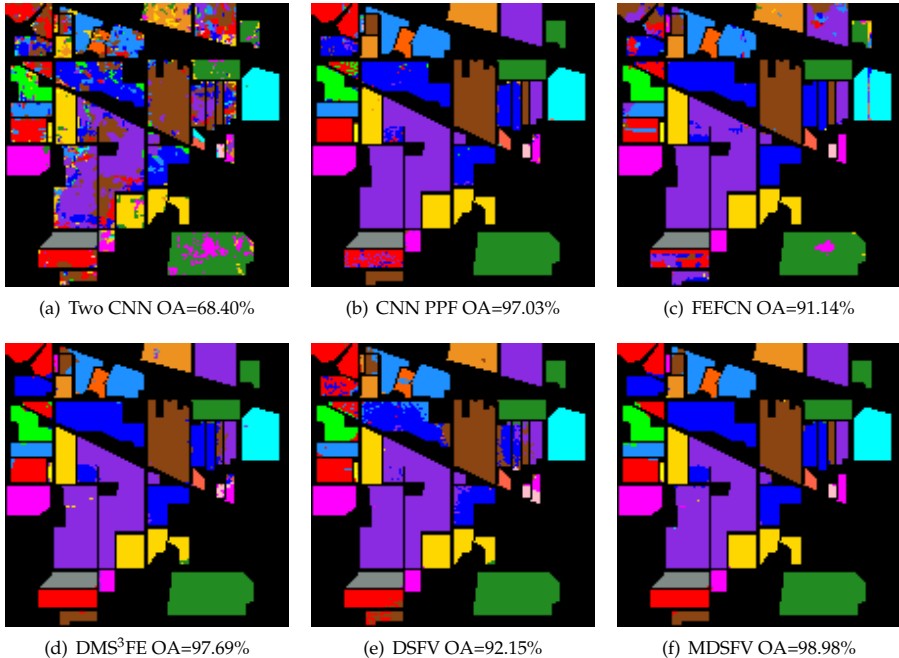

(a) Two CNN OA=68.40%　　　(b) CNN PPF OA=97.03%　　　(c) FEFCN OA=91.14%

(d) DMS³FE OA=97.69%　　　(e) DSFV OA=92.15%　　　(f) MDSFV OA=98.98%

**Figure 10.** Classification maps by compared methods on Indian Pines, the overall accuracy (OA) of the methods are displayed.

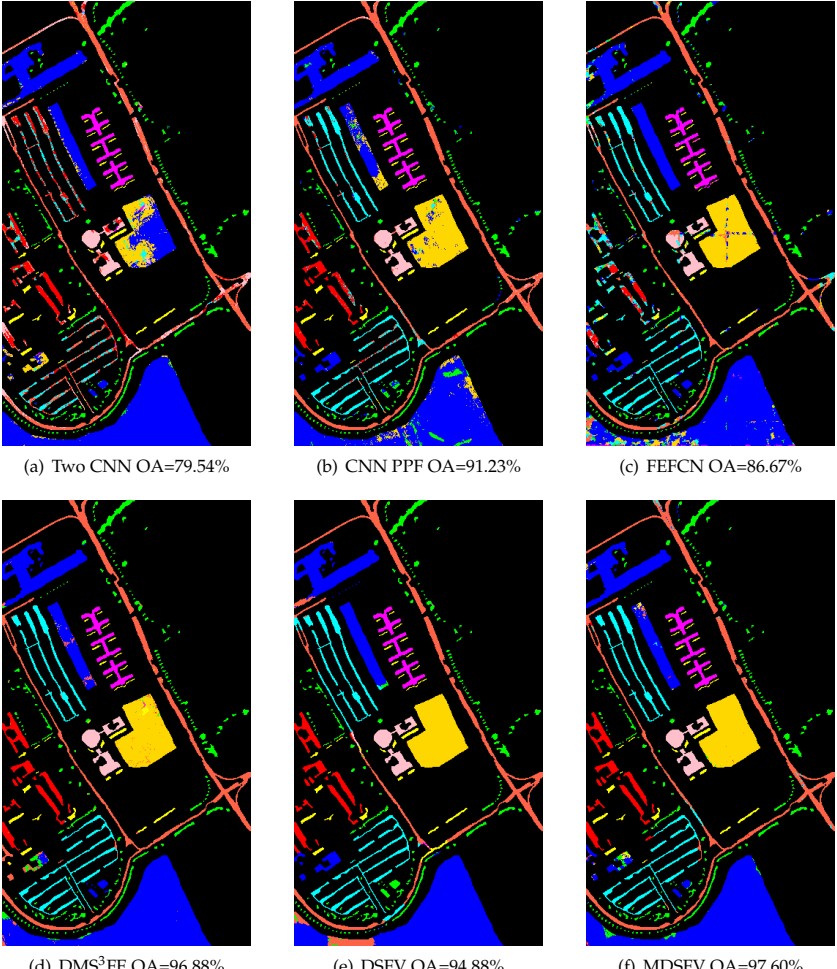

(a) Two CNN OA=79.54%　　　(b) CNN PPF OA=91.23%　　　(c) FEFCN OA=86.67%

(d) DMS³FE OA=96.88%　　　(e) DSFV OA=94.88%　　　(f) MDSFV OA=97.60%

**Figure 11.** Classification maps by compared methods on Pavia University, OA of the methods are listed.

Figure 11 and Table 8 show the classification result of various methods on the Pavia University dataset. From Figure 11, it can be seen that except for the MDSFV method in this paper, the other methods have serious segmentation errors in the meadows. It should be noted, the result of the deepest feature upsampling (DSFV) is inferior to that of DMS$^3$FE proposed in [47]. This may be due to the upsampling multiple of the spatial feature is 32, which means that the feature is inaccurate within the range of 32 pixels. But in general, our method performs well in terms of accuracy to the existing deep learning methods. Table 8 shows the accuracies per class. In the class which has major samples, MDSFV performances significantly better than the others. In terms of overall accuracy, it exceeds the DMS$^3$FE method by 0.7 percent.

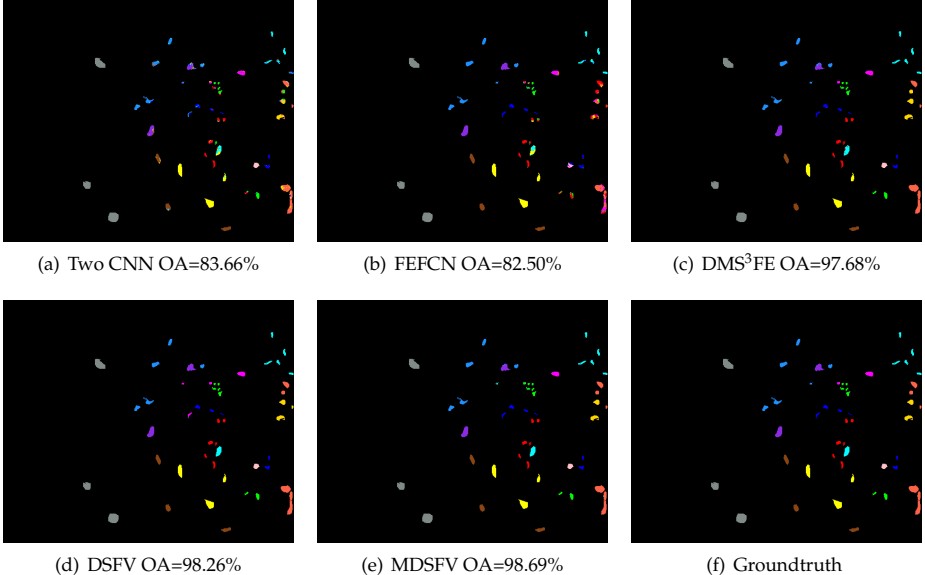

(a) Two CNN OA=83.66%       (b) FEFCN OA=82.50%       (c) DMS$^3$FE OA=97.68%

(d) DSFV OA=98.26%       (e) MDSFV OA=98.69%       (f) Groundtruth

**Figure 12.** Classification maps by compared methods on the Kennedy Space Center dataset, overall accuracy (OA)s are listed.

**Table 9.** Accuracies of different methods on the Kennedy Space Center dataset (%).

| Class | Train Sample | Test Sample | Two CNN | FEFCN | DMSFE | DSFV | MDSFV |
|-------|-------------|-------------|---------|-------|-------|------|-------|
| 1 | 20 | 741 | 79.34 ± 2.53 | 84.66 ± 4.31 | 96.84 ± 1.99 | 95.29 ± 6.97 | **96.79 ± 2.47** |
| 2 | 20 | 223 | 76.41 ± 3.46 | 77.51 ± 11.59 | 98.65 ± 2 | 96.82 ± 5.85 | **99.1 ± 0.8** |
| 3 | 20 | 236 | 79.19 ± 2.53 | 42.12 ± 11.26 | 99.45 ± 0.19 | **99.92 ± 0.17** | 99.36 ± 0.28 |
| 4 | 20 | 232 | 68.79 ± 4.39 | 60.74 ± 9.55 | 97.37 ± 2.95 | 97.54 ± 6.95 | **98.36 ± 0.54** |
| 5 | 20 | 141 | 83.33 ± 5.09 | 61.11 ± 15.06 | 93.26 ± 8.76 | **99.86 ± 0.43** | 97.45 ± 5.81 |
| 6 | 20 | 209 | 64.21 ± 7.14 | 45.76 ± 20.15 | **100 ± 0** | 99.95 ± 0.14 | **100 ± 0** |
| 7 | 20 | 85 | 98.12 ± 1.68 | 68.34 ± 25.23 | **100 ± 0** | 93.29 ± 15.7 | **100 ± 0** |
| 8 | 20 | 411 | 65.52 ± 3.9 | 97.16 ± 2.13 | 92.82 ± 5.54 | **98.83 ± 2.82** | 95.67 ± 2.23 |
| 9 | 20 | 500 | 95.44 ± 1.36 | 83.12 ± 2.27 | 97.88 ± 4.97 | **100 ± 0** | 99.74 ± 0.32 |
| 10 | 20 | 384 | 77.01 ± 10.43 | 97.5 ± 1.93 | 98.7 ± 2.9 | **100 ± 0** | 99.87 ± 0.39 |
| 11 | 20 | 399 | 81.9 ± 3.64 | **100 ± 0** | 98.05 ± 2.43 | 99.9 ± 0.12 | 96.19 ± 3.22 |
| 12 | 20 | 483 | 78.16 ± 3.37 | 98.46 ± 0.8 | 97.47 ± 3.75 | 96.4 ± 0.69 | **99.5 ± 0.7** |
| 13 | 20 | 907 | **100 ± 0** | 99.99 ± 0.03 | 99.58 ± 0.84 | 98.37 ± 4.9 | 99.88 ± 0.19 |
| | overall accuracy(OA) | | 82.68 ± 1.47 | 82.29 ± 3.87 | 97.73 ± 1.13 | 98.16 ± 1.36 | **98.53 ± 0.69** |
| | average accuracy(AA) | | 80.57 ± 2.09 | 78.19 ± 4.55 | 97.7 ± 1.19 | 98.17 ± 1.69 | **98.61 ± 0.72** |
| | Kappa | | 0.8097 ± 0.016 | 0.8039 ± 0.0419 | 0.9748 ± 0.0125 | 0.9795 ± 0.015 | **0.9836 ± 0.0077** |

Figure 12 and Table 9 demonstrate the results of various comparison algorithms on the Kennedy Space Center dataset. The CNN PPF introduced by [54] cannot obtain convergence on this dataset, even though we tried a lot of ways, including changing the training sample numbers. It may be caused by the large correlation between the spectral curves of various types of objects in the dataset, resulting in poor separability, and the accurate classification cannot be performed after the pairing algorithm. In Figure 12 when the skip-layer multiscale spatial features were not introduced, there were many mis-segments of a region. After it was introduced, the mis-classification vanished and

the phenomenon of scattered mistakes was reduced. The accuracies of classes are shown in Table 9, the overall accuracy has a 0.8% raise when compared to other state-of-art methods. Based on the above results, we can prove that the proposed skip-layer fusion of multiscale spatial features is very effective for accurate classification. The proposed unification of dimensions by PCA and normalization before the combination of features can effectively reduce the dimension of the combined features and improve the separability of features. The virtual RGB image can better fit the imaging conditions of natural images, and is more conducive to the training of the model to extract spatial features. Compared with the simple and crude three principal components, it can effectively express the pixel–spatial relationship in hyperspectral images.

## 4. Discussion

In this section, we carry out parameter analysis. We experimented with the sample size and the spatial spectral feature fusion dimension. For the sample size, we conducted experiments on the three datasets, and selected the sample numbers according to the actual situation of each dataset. For the spatial spectral feature fusion dimension, we experimented with three datasets and found for each dataset, the optimal spatial and spectral fusion feature dimensions are basically invariant, so the results of each dataset are integrated, thus, the selections of spatial and spectral feature dimension are unified.

**Sample numbers**

The number of training samples will affect the accuracy of classification. The more training samples, the higher the accuracy. However, because the sample size is limited and the time cost of training the classifier is considered, the number of samples should be appropriate.

Since the Indian pines dataset is very uneven between each class, the minimum number of samples is only 20, so we selected the training samples proportionally. The blue line in Figure 13 shows the accuracy varies with the proportion of samples on Indian Pines. We can see that the overall accuracy shows an upward trend as the number of samples increases. However, when the sample ratio reaches 10%, the overall accuracy reaches about 98%, and as the number of samples increases, the accuracy increases no longer. We determine the final sample size by 10% of each sample.

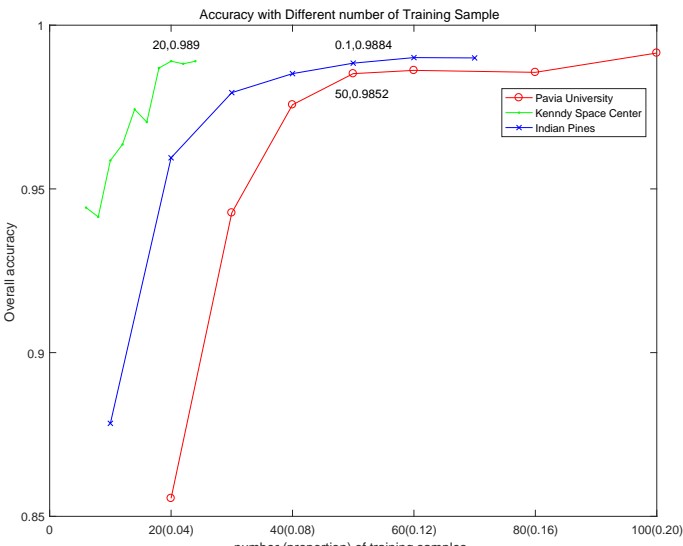

**Figure 13.** Accuracy with different numbers of training samples on the three datasets.

For the Pavia University dataset, there are nine types of samples, and the number of samples per class is relatively large, so we select according to the number of samples. The red line in Figure 13 shows the overall accuracy varies from the number of training samples each class on the Pavia University dataset. It can be obtained that when the number of samples per class is 50, the classification accuracy

reaches a level close to 98%, and the accuracy does not change significantly with the increase of the number of samples, so the number of samples is finally determined to be 50 per class.

For the KSC dataset, the number of samples per class is small and relatively balanced, so we select the number of samples per class, and the overall accuracy of classification varies with the number of samples per class as shown in Figure 13 by the green line. It can be seen that the initial improvement of the classification accuracy with the training samples is very obvious. When the number of samples per class is 20, as the number of samples increases, the classification accuracy rate increases slowly, so the final sample size is 20 per class.

### Dimension of features

In the process of spectral–spatial feature fusion, the spectral features are obtained from the original spectral PCA, and the spatial features are obtained by combining the first few dimensions of the principal components after the spatial feature PCA. The dimensions of both can be changed, i.e., in Equation (7) $s_a$, $s_e$ is changeable. We performed the experiments on three datasets and combined the results of them to obtain the final spatial and spectral feature dimension. The results on the datasets are shown in Figure 14.

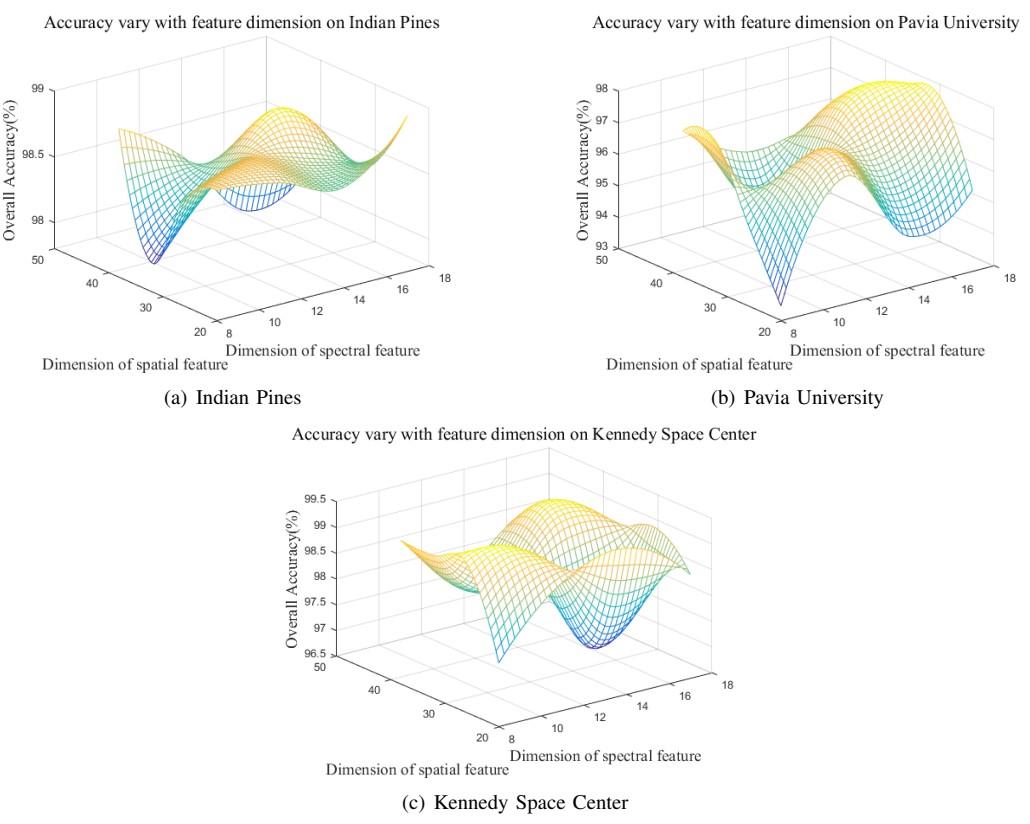

**Figure 14.** Accuracy varies with feature dimension; the spatial feature dimension varies from 24 every 6 to 42; the spectral feature dimension varies from 9 every 3 to 18; the surfaces are smoothed.

We change the spatial feature dimension from 24 every 6 dimensions to 42 and the spectral feature dimension from 9 every 3 dimensions to 18. In Figure 14 we can find that in the Kennedy Space Center dataset and Pavia University dataset, when the dimension of the spatial feature equals to 36 and in the mean time the spectral feature's dimension is 15, the accuracy reaches a local maximum, and the accuracy in the test area is almost scarcely greater than the local maximum. However, in the Indian Pines dataset, the maximum accuracy does not appear in the same place. Fortunately, the accuracy in the same place is not far from the maximum (98.83% to 98.87%), thus, we can set the dimension on all datasets as 36 for spatial feature and 15 for spectral feature. We can find that for the three datasets for experiments, when the spatial and spectral feature dimensions are within a reasonable

range, the classification accuracy does not change significantly. This proves that the parameters are allowed in this range and our spatial and spectral feature dimensions are robust to some extent.

In this section, we discussed the accuracy varies with the training sample numbers. Meanwhile, relevant parameters of feature dimension is selected in the designed experiment.

## 5. Conclusions

We propose a hyperspectral classification method based on multiscale spatial feature fusion. We introduce a new three-channel image combination method to obtain virtual RGB images. In these images, the hyperspectral corresponding bands are synthesized by simulating the RGB imaging mechanism of natural images. The image is used to extract multiscale and multi-level spatial features in a network trained on natural images, which can better fit the model parameters trained on natural images and obtain more effective spatial features. By combining the multiscale spatial features, the semantic information of the deep features can be utilized simultaneously to ensure the accuracy of the feature classification and the edge detail information of the shallow features can ensure the regularity and continuity of the edge classification of the feature. The proposed skip-layer feature combination method can avoid the problem that the feature dimension increases which is caused by the traditional concatenation method, the long time-consumption in classification and the separability decreases. Experiments showed that our method performs well compared to the previous deep learning methods and achieves a higher classification accuracy rate. In a future work, we will further study the time performance and complexity of the algorithm. In addition, the virtual RGB image we introduced provides a new solution for all algorithms involving the synthesis of three-channel images. This solution can avoid PCA's simple and crude reduction of data, which can better adapt to the characteristics of the deep learning networks on natural images. It can bridge the gap between hyperspectral data and natural images.

**Author Contributions:** conceptualization, Z.S. and L.L.; methodology, L.L.; validation, L.L. and B.P.; formal analysis, N.Z., H.L. and X.L.; writing—original draft preparation, L.L.; writing—review and editing, L.L., Z.S. and B.P. All authors have read and agreed to the published version of the manuscript.

**Funding:** This work was supported by the National Key R&D Program of China under the Grant 2017YFC1405605, the National Natural Science Foundation of China under the Grant 61671037, the Beijing Natural Science Foundation under the Grant 4192034 and Shanghai Association for Science and Technology under the Grant SAST2018096.

**Conflicts of Interest:** The authors declare no conflict of interest.

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
