# Peer review of "Multiscale Deep Spatial Feature Extraction Using Virtual RGB Image for Hyperspectral Imagery Classification"

_remotesensing, doi:10.3390/rs12020280_

Round 1

Reviewer 1 Report

In this paper the authors present a method for the classification of hyperspectral images based on fully neural networks.

The contribution consists of the creation of a virtual RGB image and a multiscale fusion method combining spectral and spatial features.

The paper is organized in a proper fashion.

The manuscript needs to be polished for the sake of clarity of meaning (see below comments).

Major comments:

The authors make a lot of unexplained assumptions that are not explained:

Why did the choose pool3, pool4 and fc7 layers?

At line 177 they affirm  “The method mainly through the upsampling, cropping and two

layer feature maps joint to realize layer-by-layer joint thus improve the ability of expressing spatial charicteristics”. This claim is not supported by any evidence.

At line 190 they say: “take the principal component of the same dimension as that of the low feature map” but principal component dimension depends on the data dimension. Maybe they wanted to say that they take a number of principal components equal to the dimension of the low feature map, but it is not clear.

At line 243 the authors talk of “one-to-one  correspondence with each pixel of the original image” how this is possible with all the downsampling and upsampling that are involved in the procedure?

As far as I understood in Section 3.1 they split the dataset in 10% for the training and 90% for the test step. Why did they make this (not common) choice?

At line 361 the authors say “Fig. 7 shows the effctiveness of normalization, it can be find the result is closer to  groundtruth.” Please give some quantitive evaluation, from Fig.7 it is not possible to conclude anything.

It is not clear what data are shown in the tables. Accuracy? How is it computed?

Why in Table 5 there are 100 samples in the training phase regardless the number of samples in the test phase?

Why the data are divided by classes? I thought that images, containing pixels with different labels (classes), were the input of the algorithm. If the authors divided the training and test by classes, it seems that the perform the analysis using pixels and not the full images.

Minor comments:

Page 2 line 44: “However, the effect is not satisfying thus cannot meet the requirement of accuracy on ground object

classification using a certain number of sample” It is not clear what “effect” the authors are talking about. Please rephrase

Page 2 line 64: “These networks artificially construct a data volume for each pixel to be  classified as an input, actually it is the process to obtain the deep features to classify from individual  shallow features.” This sentence it is not clear. What do the authors mean with “actually it is the process to obtain the deep features to classify from individual  shallow features”? Do they mean that the construction of a data volume creates deep features from shallow ones?

Page 4 line 138: “many scholars have used the well-trained networks on natural images to extract  features from hyperspectral imag” What is this well-trained network(s)? Please specify

Page 4 line 141: “and then select the first three principal components to form a three-channel  image into the networks” What does mean “form a image into the network” ? How the first three PCA components are weighted to form a three-channel  image

Page 5 line 164: “Take R band” please uniform the capitol case in equations 1-4

Page 5 line 165: “The advantages of FCN to extract the spatial features is that the same  target with hyperspectral classification” I am sorry but I do not get what this sentence means.

Page 5 line 181: “Since the shallow feature map occurred after layer-to-layer convolution from the zero padded original image.” The verb is missing

Page 7 line 195: “w h is the size of feature to dimension reduce.” What does this mean?

Page 7 line 206: “before and after” before and after what?

Page 8 line 217:  “if the shallower’s dimension is more, PCA will be also used on it.” More what?

Page 10 line 301: “Since each type of labeled sample has large amount of samples,” I am sorry but I am lost with sample having samples

Page 11 line 312: “caffe framework”, please add an appropriate reference for Caffe

Page 14 line 362: “Including PCA the hyperspectral data directly” what does it mean?

Page 17 figure 11: this figure is too small.

Please add in the caption of the figures and tables the dataset they refer to.

Author Response

RESPONSES TO THE REVIEWER 1

We appreciate it very much for your time and your very insightful comments on our manuscript. We have given a detailed response to all the concerns that you addressed. Important modifications are highlighted in the one column manuscript by red font. We hope that our modification could improve the quality of this manuscript.

Specially, we separated the results and discussion section into results section and discussion section respectively at the request of the editor. We carefully checked the references and figures and tables in our revision. The detailed revision of the manuscript is in our detailed response as follows.

Best wishes and thanks again.

=============================================================

An item-by-item response to all the problems the reviewer concerns are shown as follows. We highlight important modifications in the one column manuscript, and the locations mentioned below also correspond to the one column manuscript.

Reviewer’s overall comments:

In this paper the authors present a method for the classification of hyperspectral images based on fully neural networks.

The contribution consists of the creation of a virtual RGB image and a multiscale fusion method combining spectral and spatial features.

The paper is organized in a proper fashion.

The manuscript needs to be polished for the sake of clarity of meaning (see below comments).

Response to overall comments:

Thank you for your deep understanding about our work. In the resubmitted manuscript, We have revised the unclear part of the expression in detail. We defined the description of the feature extraction and PCA processes in Section 2 (Page 4-9). Meanwhile, we enriched the caption of the figures and tables in Section 3 (Page 10-17).

We hope that our revise could improve the quality of this work. Many thanks again for your careful review.

Comment 1: The authors make a lot of unexplained assumptions that are not explained: Why did the choose pool3, pool4 and fc7 layers?

Response to 1: Thank you for your very meaningful comments. We have lack of explanation on the choose of feature layers. Here we explain it as follows:

During the pooling operation of the Fully Convolution Network (FCN), the down-sampling multiples of spatial features increase gradually and the semantic properties of features are more and more abstract. Deep features can provide more semantic information to facilitate the classification of pixel categories, while shallow features can provide more edge detail information to facilitate the classification of objects edges.

In the network we used to extract features, fc7 had the highest degree of feature abstraction, pool5 was the same in the down-sampling scale with it, so we used fc7 for spatial feature joint, which only after two convolution layers from pool5. So we pay more attention to other layers which have different semantic character characteristic with fc7 such as pool3 and pool4, ignoring pool5.

Actually, selecting which layer feature to use for spatial feature fusion is not must be fixed. The key of our method is to select features of different semantic levels for jointing, so that spatial features contain both semantic information and detailed information.

In the Feature Jointing And Fusion Strategies subsection in Results section, we compared the performance of different depth of spatial feature in figure 6, according to the experiment, fusion of pool3, pool4 and fc7 occurred a best performance.

That's why we chose fc7 and pool3,pool4 for feature jointing.

In the revised manuscript, to avoid misunderstanding about the above problem, we supplement the following description at Line 167.

“During the pooling operation of the Fully Convolution Network (FCN), the down-sampling multiples of spatial features increase gradually and the semantic properties of features are more and more abstract. The fc7 layer provides semantic information and the shallower layers provide more detailed information. So we choose both deep and shallow features, we detailed extracted the features of pool3, pool4 and fc7 layers and combined them.”

Comment 2: At line 177 they affirm “The method mainly through the upsampling, cropping and two layer feature maps joint to realize layer-by-layer joint thus improve the ability of expressing spatial characteristics”. This claim is not supported by any evidence.

Response to 2: Thank you for your careful comments. We expressed not clearly here. We said that the method can improve the ability of expressing spatial characteristics is supported by the description after that sentence. The method combine both semantic information in fc7 and the edge texture information of the shallow features.

To make the description more logically progressive, we changed the sentence into ”The method mainly through the upsampling, cropping and two layer feature maps joint to realize layer-by-layer joint expecting improve the ability of expressing spatial characteristics. It can better preserve the depth semantic information in the fc7 layer, and simultaneously combine the edge texture information of the shallow features to improve the ability of feature expression.” in line 174.

Comment 3: At line 190 they say: “take the principal component of the same dimension as that of the low feature map” but principal component dimension depends on the data dimension. Maybe they wanted to say that they take a number of principal components equal to the dimension of the low feature map, but it is not clear.

Response to 3: Thank you for your insightful comments. We lack description on this part. Actually, we want to tell the readers that when the size of feature map is small, even the dimension is very large (assumed n), according to the principle of PCA, the dimension after PCA is (w*h-1), if the other spatial feature (assumed n) less than (n) but more than (w*h-1), that is , we select the dimension of the two layers both (w*h-1) rather than (m).

In the revised manuscript, we give a more clear express in line 197 and 200 as:

“Here we use principal component analysis (PCA) reducing the dimension of the feature map with high dimension to make the dimension of the two maps same and add them layer by layer.”

“In the process of dimensionality reduction, by the characteristics of the PCA itself, our available principal components are less than , is the size of feature to PCA. So if the dimension of the shallower is  and that of the deeper is , the size of the deeper map is , the dimension of the two maps is ”.

Comment 4: At line 243 the authors talk of “one-to-one  correspondence with each pixel of the original image” how this is possible with all the downsampling and upsampling that are involved in the procedure?

Response to 4: Thank you for your kind correction. You pointed out clearly that we had a representation error, the spatial feature is not one-to-one corresponding with the image. And in fact it depends on our lower sampling multiple, in the method we use, it has a 8 pixel deviation. Thank you again for your serious and rigorous comments.

In the manuscript, we changed the description in line 252 to:“The feature map corresponds to the original image as much as possible, which can effectively reduce the possibility of generating classification errors in the two types of handover positions.”

Comment 5: As far as I understood in Section 3.1 they split the dataset in 10% for the training and 90% for the test step. Why did they make this (not common) choice?

Response to 5: Thank you for your meaningful question. The split of the training and testing samples is different for the three datasets we use. If the numbers of the classes is equally large, we choose the training sample of each class a definite number. Otherwise, we select training numbers according to the proportion of each type of sample. In Indian pines dataset, we select 10% of the samples because the number of different class is severe inequality.

In order to simplify the problem, we did not specify different rules for the number of samples of different categories, because we compared the algorithms under the same number of samples, so the comparison was fair. The problem raised by the reviewer is useful and we will pay more attention to it in the next works.

Comment 6: At line 361 the authors say “Fig. 7 shows the effctiveness of normalization, it can be find the result is closer to groundtruth.” Please give some quantitive evaluation, from Fig.7 it is not possible to conclude anything.

Response to 6: Thank you for your careful comments. Indeed, the difference between the two methods cannot be clearly seen on the figure 7 due to the small difference in accuracy. And the quantitive evaluation is showed in Table 4 in Page 13.

So for the sake of describing rigor, we revised the sentence in line 371 into “Fig. 7 shows the result figures with and without normalization.”

Comment 7: It is not clear what data are shown in the tables. Accuracy? How is it computed?

Response to 7: Thank you for your rigorous question. The accuracy shown in the tables means the accuracy of every class, which is obtained by dividing the samples correctly classified by the total samples of the class. As the following formula shows:

 represents the accuracy of the  class,  represents the number of truly classed samples in  class,  is the sample number of  class of test sample.

In addition, the indicators to measure the performance of all categories are the overall accuracy (OA), the average accuracy (AA) and the kappa coefficient (). Since them are widely used in the evaluation of hyperspectral classification methods, we did not give formulaic description of them. They are calculated by the following formula:

Where,  represents the number of samples correctly classified, and  represents the total number of samples tested.

Average accuracy (AA) is the percentage of pixels each class correctly classifies at the class level

Where,  represents the number of classification categories,  represents the total number of test samples in the  category, and  represents the number of correct classification in the  category. It can be seen that the average accuracy is the average accuracy of each category.

The kappa coefficient () is a measure of the randomness of the correct classification of samples

Where,  is the sum of the samples of each correct classification divided by the total samples, that is, the overall classification accuracy, .

is calculated from the following formula

Where,  is the total test sample number, , is the sample number of each type of test sample, and  is the sample number of each type of correct classification.

In order to make readers have a clearer understanding, we have made the following modifications in the manuscript in line 331.

“and the accuracies of each class shown in the tables are calculated by the proportion of correctly classified samples to total test samples.”

Comment 8: Why in Table 5 there are 100 samples in the training phase regardless the number of samples in the test phase?

Response to 8: Thank you for your meaningful questions. In the subsection 3.2, we compared the effectiveness of our strategy in the feature joint and fusion, the training samples of each class is 100. The sample number is not consistent with the comparison experiments with other algorithms in subsection 3.4. That is because in the algorithm design phase, we just compare the different strategies on same conditions. As long as the number of samples is the same, the validity analysis of the strategies can be given by comparing the accuracies.

In the comparison with other algorithms, we changed the number of samples to reflect the gap between our method and others. If the sample numbers is 100, the accuracies is more closer to 100%, the gap between the algorithms will be smaller, inconvenient to compare. So that we change the numbers of each class to 50 in the comparison.

Comment 9: Why the data are divided by classes? I thought that images, containing pixels with different labels (classes), were the input of the algorithm. If the authors divided the training and test by classes, it seems that the perform the analysis using pixels and not the full images.

Response to 9: Thank you for your insightful comments. This is really a problem in the field of HSI classification. Since the the imaging conditions of different hyperspectral imagery, the number of spectral bands and the ground objects are significantly different, which make different hyperspectral datas can not be processed together like natural images and other remote sensing images. Researchers always divide the training set and test set on the same imagery.

In addition, our work focuses on the algorithm research. In this case, we have to use public data sets in the same way with other published papers so that the comparison experiments are relatively fair.

Actually, a research of a method you mentioned which divide the training samples by different images (some images to train and others for test) is very meaningful. This is based on a method which can unify all hyperspectral imagery into the same spectral dimension and style. That's what we are aiming for next!

Comment 10: Page 2 line 44: “However, the effect is not satisfying thus cannot meet the requirement of accuracy on ground object classification using a certain number of sample” It is not clear what “effect” the authors are talking about. Please rephrase.

Response to 10: Thank you for your careful comments. To make it clear, we revised it into “However, the methods cannot meet the requirement of accuracy on the condition that samples is very limited. ” in line 44.

Comment 11: Page 2 line 64: “These networks artificially construct a data volume for each pixel to be classified as an input, actually it is the process to obtain the deep features to classify from individual shallow features.” This sentence it is not clear. What do the authors mean with “actually it is the process to obtain the deep features to classify from individual shallow features”? Do they mean that the construction of a data volume creates deep features from shallow ones?

Response to 11: Thank you for your kind comments. In this sentence, we want to express that the methods construct a data volume for each pixel as input. This input is a simplest shallow feature and the method turned into the process extracting deep features from shallow features (the input).

To reduce ambiguity, we revised the manuscript in line 65 into “This input can be seen as the simplest shallow features and the method actually turned into a process to obtain the deep features from the input shallow features.”

Comment 12: Page 4 line 138: “many scholars have used the well-trained networks on natural images to extract features from hyperspectral images” What is this well-trained network(s)? Please specify

Response to 12: Thank you for your careful suggestions. The well trained networks used is mainly CNN and FCN. In the revised manuscript, we detailed the networks and related references.

In line 129 “In recent years, many scholars have used the well-trained networks on natural images to extract features from hyperspectral images, such as CNN [50] and FCN [47].”

Comment 13: Page 4 line 141: “and then select the first three principal components to form a three-channel image into the networks” What does mean “form a image into the network”? How the first three PCA components are weighted to form a three-channel image.

Response to 13: Thank you for your meaningful question. The three components are normalized to 0-255 respectively, then we simply combine the first, second, third component as the red, green and blue channel to three-channel image. It is mainly as the following figure:

In the manuscript, since the process is described by the [47], we didn’t give detailed describe about that. we revised the sentence in line 132 into “detailed process can be find in [47].”

Comment 14: Page 5 line 164: “Take R band” please uniform the capitol case in equations 1-4

Response to 14: Thank you for your careful comment. We have carefully revised equations 1-4 from “r” to “R”. You can find them in Page 5.

Comment 15: Page 5 line 165: “The advantages of FCN to extract the spatial features is that the same target with hyperspectral classification” I am sorry but I do not get what this sentence means.

Response to 15: Thank you for your kind comments. In this sentence, we want to say why we chose FCN to extract the spatial feature. The advantage of FCN is that it have same target with the hyperspectral image classification, aiming at pixel-wise classification. We reasonably guess that compare to CNN, features from FCN is more useful.

To make the describe clear, we revised it in line 156:

“We select FCN for feature extraction. The advantage of FCN is that it have same target with the hyperspectral image classification, aiming at pixel-wise classification. We reasonably guess that compare to CNN, features from FCN is more useful.”

Comment 16: Page 5 line 181: “Since the shallow feature map occurred after layer-to-layer convolution from the zero padded original image.” The verb is missing

Response to 16: Thank you for your meaningful correction. In the revised manuscript, we correct it to: “The shallow feature maps are occurred after layer-to-layer convolution from the zero padded original image.” in line 188.

Comment 17: Page 7 line 195: “w×h is the size of feature to dimension reduce.” What does this mean?

Response to 17: Thank you for your meaningful questions. We have explicitly specified that w×h represents the size of the deeper feature in the revised manuscript, you can find it in line 202.

“So if the dimension of the shallower is  and that of the deeper is , the size of the deeper map is , the dimension of the two maps is ”.

Comment 18: Page 7 line 206: “before and after” before and after what?

Response to 18: Thank you for your good questions. To make it clear we revised the manuscript in line 215 to: “We discuss the calculation of the offset between the two layers of feature maps before and after the pooling and other convolution operations.”

Comment 19: Page 8 line 217: “if the shallower’s dimension is more, PCA will be also used on it.” More what?

Response to 19: Thank you for your kind questions. As we said in line 200, if the dimension of the shallower feature (m), (m)>(w×h-1), the dimension we use is (w×h-1), so we also do PCA on the shallower features to change its dimension from (m) to (w×h-1).

To make it clear, we revised the sentence into “If the shallower's dimension is the larger, PCA will be also used on it to change the dimension to that of the deeper feature after PCA.” in line 226.

Comment 20: Page 10 line 301: “Since each type of labeled sample has large amount of samples,” I am sorry but I am lost with sample having samples.

Response to 20: Thank you for your careful comments. We have a small error here. We correct it into: “Since each type of labeled sample has a large amount” in line 310.

Comment 21: Page 11 line 312: “caffe framework”, please add an appropriate reference for Caffe.

Response to 21: Thank you for your meaningful suggestions. We carefully cited the

[51]Jia, Yangqing, et al. "Caffe: Convolutional architecture for fast feature embedding." Proceedings of the 22nd ACM international conference on Multimedia. ACM, 2014.

in line 322 in the revised manuscript as [51].

Comment 22: Page 14 line 362: “Including PCA the hyperspectral data directly” what does it mean?

Response to 22: Thank you for your kind comments. In this subsection, we compared accuracy of different ways to combine three-channel image. PCA the hyperspectral data directly, the average of RGB corresponding bands and Gaussian combination of the bands are compared. We have some ambiguity in the presentation.

So we revised the line 392 into: “PCA the hyperspectral data directly, the average of RGB corresponding bands and Gaussian combination of the bands are compared.” to clear it.

Comment 23: Page 17 figure 11: this figure is too small.

Response to 23: Thank you for your careful remind. We revised the size of the Figure 14 in Page 20.

Comment 24: Please add in the caption of the figures and tables the dataset they refer to.

Response to 24: Thank you for your kind comments. We have added the dataset the figures and tables refer to in their caption. You can see that in Figure 6-9 and 13, Table 3-6.

Reviewer 2 Report

The paper addresses an interesting approach to hyper-spectral imagery classification. The authors propose an improved hyper-spectral classification method based on multi-scale spatial feature fusion, and introduced a new three-channel image combination method to obtain virtual RGB images with the hyperspectral corresponding bands being synthesized by simulating the RGB imaging mechanism of natural images. Such generated images are used to extract multi-scale and multi-level spatial features in a network trained on natural images. Furthermore,the combining of the multi-scale spatial features is used. The authors proposed cross-layer feature combination method in order to avoid the problem of increasing feature dimension. The performance of the proposed MDSFV solution are compared with other previously proposed deep learning methods, and it is shown that it achieves a higher classification accuracy rate.

The manuscript is well devised and written, with relatively high presentation and technical level. However, some minor issues could be addressed:

I. In section 3.5 Classification performance, a performance comparison for the proposed and 4 other previously known solutions is given. The performance estimation is performed by using the parameters and sample size that are fitted for the here proposed MDSFV solution (section 3.4). Some additional remarks could be given in order to explain how these parameter choices influence other solutions used in performance comparison (are the chosen values appropriate and the underlying justification if not). Also, the solutions are also compared only by the means of classification success performance - a complexity and time consumption data could also be very interesting.

II. Some sentences are unclear and could be rephrased, i.e.: (1) The first sentence in paragraph starting in line 177, (2) sentence in lines 249-250 (double verb?), (3) sentence in lines 257-260 is unnaturally splitted, (4) part of paragraph in lines 288-293, (5) the recurring phrase given in lines 294-295, 302-303, 310-311, (6) sentence in lines 371-372,...

III. Some language errors could be corrected, i.e.: (1) word datas is used several times instead of data, e.g. line 123, (2) the expression cross-layer jointing is somewhat inappropriate, ... The manuscript as a whole can be revisited and all spelling and other errors corrected.

Author Response

RESPONSES TO THE REVIEWER 2

We appreciate it very much for your time and your very insightful comments on our manuscript. We have given a detailed response to all the concerns that you addressed. Important modifications are highlighted in the one column manuscript by red font. We hope that our modification could improve the quality of this manuscript.

Specially, we separated the results and discussion section into results section and discussion section respectively at the request of the editor. We carefully checked the references and figures and tables in our revision. The detailed revision of the manuscript is in our detailed response as follows.

Best wishes and thanks again.

=============================================================

An item-by-item response to all the problems the reviewer concerns are shown as follows. We highlight important modifications in the one column manuscript, and the locations mentioned below also correspond to the one column manuscript.

Reviewer’s overall comments:

The paper addresses an interesting approach to hyper-spectral imagery classification. The authors propose an improved hyper-spectral classification method based on multi-scale spatial feature fusion, and introduced a new three-channel image combination method to obtain virtual RGB images with the hyperspectral corresponding bands being synthesized by simulating the RGB imaging mechanism of natural images. Such generated images are used to extract multi-scale and multi-level spatial features in a network trained on natural images. Furthermore,the combining of the multi-scale spatial features is used. The authors proposed cross-layer feature combination method in order to avoid the problem of increasing feature dimension. The performance of the proposed MDSFV solution are compared with other previously proposed deep learning methods, and it is shown that it achieves a higher classification accuracy rate.

The manuscript is well devised and written, with relatively high presentation and technical level. However, some minor issues could be addressed:

Response to overall comments:

Thank you for your deep understanding about our work. We have carefully revised the manuscript according to your suggestions. We defined the description of the feature extraction and PCA processes in Section 2 (Page 4-9). Meanwhile, we enriched the caption of the figures and tables in Section 3 (Page 10-17). We hope that our revise could improve the quality of this work. Many thanks again for your careful review.

Comment 1: In section 3.5 Classification performance, a performance comparison for the proposed and 4 other previously known solutions is given. The performance estimation is performed by using the parameters and sample size that are fitted for the here proposed MDSFV solution (section 3.4). Some additional remarks could be given in order to explain how these parameter choices influence other solutions used in performance comparison (are the chosen values appropriate and the underlying justification if not). Also, the solutions are also compared only by the means of classification success performance - a complexity and time consumption data could also be very interesting.

Response to 1: Thank you for your very meaningful comments. In our method, there are parameters of two aspects: one is the training sample numbers and another is the dimension of spatial and spectral features.

In case of the number of training samples, all the methods to compare have a Equivalent parameter. It can be seen under the same condition, our proposed method achieved the best result. We have reason to believe that our algorithm can still obtain better results than other algorithms under other sample sizes. This is recognized in the study of hyperspectral classification. Other previously published papers are also compared under a sample condition to judge the pros and cons of the algorithm, such as the references [2-7] of our manuscript.

In case of the dimension of the spatial and spectral features, Figure 14 in our manuscript demonstrates the overall accuracy varies with the dimensions. We can find that for the three datasets for experiments, when the spatial and spectral feature dimensions are within a reasonable range, the classification accuracy does not change significantly. This proves that the parameters are allowed in this range and our spatial and spectral feature dimensions are robust to some extent.

In our algorithm, the accuracy of the algorithm is more concerned, and the complexity and time performance of the algorithm are ignored to a certain extent. This will be a research goal for our next work. Intuitively, our algorithm does not require the training process of the model, which will improve the time performance of the algorithm to a certain extent. We will also conduct quantitative research in the next step.

To make the description of the parameters’ analysis more clear, we revised the discussion section. In line 487, we added description as follows:

“We can find that for the three datasets for experiments, when the spatial and spectral feature dimensions are within a reasonable range, the classification accuracy does not change significantly. This proves that the parameters are allowed in this range and our spatial and spectral feature dimensions are robust to some extent.”

To illustrate the focus of our approach and the direction of future efforts, I added a note to the conclusion in line 506 as following:

“In the future work, we will further study the time performance and complexity of the algorithm.”

Comment 2: Some sentences are unclear and could be rephrased, i.e.: (1) The first sentence in paragraph starting in line 177, (2) sentence in lines 249-250 (double verb?), (3) sentence in lines 257-260 is unnaturally splitted, (4) part of paragraph in lines 288-293, (5) the recurring phrase given in lines 294-295, 302-303, 310-311, (6) sentence in lines 371-372,...

Response to 2: Thank you for your careful comments. We carefully revised the sentences to make them clear.

We expressed not clearly on the feature extraction, to make the description more logically progressive, we changed the sentence into “The method mainly through the upsampling, cropping and two layer feature maps joint to realize layer-by-layer joint expecting improve the ability of expressing spatial characteristics. It can better preserve the depth semantic information in the fc7 layer, and simultaneously combine the edge texture information of the shallow features to improve the ability of feature expression.”in line 174. The sentence “We use the hyperspectral bands of RGB-corresponding wavelengths to combine to construct a virtual RGB image.”is not distinct. We revised it into “We combine the hyperspectral bands of RGB-corresponding wavelengths to construct a virtual RGB image, and then use FCN to extract multi-layer, multiscale features of the image.” in line 259. “To ensure that the feature dimension is not too high during the classification process and the expression ability of the feature is not affected as much as possible, we carry out the spectral curve for PCA. And select the former masters of the composition as a spectral feature of the pixel.”is splitted We revised it in line 269 into “After the PCA we select the former masters of the composition as a spectral feature of the pixel.” The sentence is not clear describing the Indian Pines dataset. We revised the sentence to “After removing the influence band of noise and water absorption ([104-108], [150-163], 220), the remaining 200 bands are used for experiments. ”in line 298. The figure of the three datasets are shown respectively in Fig 3-5. We described the content of the figures in a same way, so the sentences looked recurring. To make the language more natural, we revised them respectively as follows:

“The PCA first three components image, the virtual RGB image and the corresponding objects label map are as shown in Fig. 3.” in line 304.

“In Fig. 4, the three components image, the virtual RGB image and label of Pavia University are displayed.” in line 310.

“Fig. 5 shows the three-channel image by PCA, the virtual RGB image and the label map of KSC. ” in line 320.

There is an ambiguous description:”Fig.8exhibits the results of the three ways. It is shown that when use spectral feature only, the continuity between pixel classes is not good, and there is a phenomenon in which pixels are scattered.” We want to describe the problem of applying only spatial features. To make it clear, we revised the sentence into “Fig. 8 exhibits the results of using spectral or feature respectively and that of the fusion feature. When using spectral feature only, the continuity of the spatial distribution of ground features is affected, such as pixels in the shandow class are classified very scattered.” in line 380.

Comment 3: Some language errors could be corrected, i.e.: (1) word datas is used several times instead of data, e.g. line 123, (2) the expression cross-layer jointing is somewhat inappropriate, ... The manuscript as a whole can be revisited and all spelling and other errors corrected.

Response to 3: Thank you for your insightful comments. Sorry for the inconvenience of your review due to language errors and unclear descriptions. We carefully fixed the language and grammatical errors of all the manuscript. Such as:

“which make different hyperspectral data can not be trained together like natural images and other remote sensing images.” in line 113.

We changed the expression of the “cross-layer jointing” into “skip-layer” in the full manuscript.